# Bacterial autolysins trim cell surface peptidoglycan to prevent detection by the Drosophila innate immune system

**Magda Luciana Atilano[1,2†], Pedro Matos Pereira[3†], Filipa Vaz[1], Maria João Catalão[1], Patricia Reed[3], Inês Ramos Grilo[4], Rita Gonçalves Sobral[5], Petros Ligoxygakis[2], Mariana Gomes Pinho[3], Sérgio Raposo Filipe[1]\***

[1]Laboratory of Bacterial Cell Surfaces and Pathogenesis, Instituto de Tecnologia Química e Biológica, Universidade Nova de Lisboa (ITQB-UNL), Oeiras, Portugal; [2]Genes and Development Laboratory, Department of Biochemistry, University of Oxford, Oxford, United Kingdom; [3]Laboratory of Bacterial Cell Biology, Instituto de Tecnologia Química e Biológica, Universidade Nova de Lisboa (ITQB-UNL), Oeiras, Portugal; [4]Laboratory of Molecular Genetics, Instituto de Tecnologia Química e Biológica, Universidade Nova de Lisboa (ITQB-UNL), Oeiras, Portugal; [5]Departamento de Ciências da Vida, Centro de Recursos Microbiologicos (CREM), Faculdade de Ciências e Tecnologia, Universidade Nova de Lisboa, Caparica, Portugal

**Abstract** Bacteria have to avoid recognition by the host immune system in order to establish a successful infection. Peptidoglycan, the principal constituent of virtually all bacterial surfaces, is a specific molecular signature recognized by dedicated host receptors, present in animals and plants, which trigger an immune response. Here we report that autolysins from Gram-positive pathogenic bacteria, enzymes capable of hydrolyzing peptidoglycan, have a major role in concealing this inflammatory molecule from *Drosophila* peptidoglycan recognition proteins (PGRPs). We show that autolysins trim the outermost peptidoglycan fragments and that in their absence bacterial virulence is impaired, as PGRPs can directly recognize leftover peptidoglycan extending beyond the external layers of bacterial proteins and polysaccharides. The activity of autolysins is not restricted to the producer cells but can also alter the surface of neighboring bacteria, facilitating the survival of the entire population in the infected host.

\*For correspondence: sfilipe@itqb.unl.pt

†These authors contributed equally to this work

**Competing interests:** The authors declare that no competing interests exist.

## Introduction

Peptidoglycan (PGN) is a macromolecule composed of long glycan strands, cross-linked by short peptides, which surrounds most bacterial cells and forms a load-bearing mesh that sustains their shape. Enlargement and remodeling of the PGN mesh during bacterial growth and division requires not only the synthesis of new material, catalyzed by penicillin-binding proteins (PBPs), but also PGN hydrolysis to allow insertion of new material, carried out by PGN autolysins (*Vollmer et al., 2008*).

Despite its essential role, PGN may also be viewed as an 'Achilles heel' of bacteria during infection, as different hosts have specialized receptors that recognize PGN as a pathogen-associated molecular pattern (PAMP) and initiate an inflammatory response to eliminate invading bacteria. Examples of these receptors include LysM proteins in plants (*Willmann et al., 2011*), intracellular NOD-like receptors (NLRs) and Toll-like receptors (TLRs) in mammals (*Chaput and Boneca, 2007*), and peptidoglycan recognition proteins (PGRPs) in insects and mammals (*Dziarski and Gupta, 2006*).

PGRPs, originally isolated due to their high affinity to PGN (*Kang et al., 1998*), are used by *Drosophila* flies to distinguish between Gram-negative and Gram-positive bacteria, directly at the

**eLife digest** While most bacteria are harmless, some can cause diseases as varied as food poisoning and meningitis, so our immune system has developed various ways of detecting and eliminating bacteria and other pathogens. Receptor proteins belonging to the immune system detect molecules that give away the presence of the bacteria and trigger an immune response targeted at the invading pathogen.

Peptidoglycan is one telltale molecule that betrays the presence of bacteria. Peptidoglycan is found in the bacterial cell wall, and for many years it was assumed that the immune system detected stray fragments of peptidoglycan that were accidentally shed by the bacteria. However, it was later shown that the immune system could, under certain conditions, detect peptidoglycan when it is still part of the cell wall. This raised an interesting question: do bacteria use other methods to stop peptidoglycan being detected by the immune system?

Now, Atilano, Pereira et al. have found that enzymes called autolysins can conceal bacteria from the receptor proteins that detect peptidoglycan. These enzymes are needed to break the bonds within the peptidoglycan present in the rigid bacterial cell wall to allow the bacteria to grow and divide. 'Knocking out' the genes for autolysins allowed the receptor proteins from the fruit fly, *Drosophila*, to bind to the bacteria; however, the mutant bacteria were able to evade the immune system after they had been treated with the purified enzymes.

Atilano, Pereira et al. suggest that the autolysins trim the exposed ends of the peptidoglycan molecules on the surface of the cell wall, which could otherwise be detected by the host. The experiments also show that bacterial pathogens—including a strain of MRSA—with mutations that knock out autolysin activity trigger a stronger immune response in fruit flies, and are therefore less able to infect this host. Autolysins also help to conceal *Streptococcus pneumoniae*—a bacterial pathogen that is a common cause of pneumonia and infant deaths in developing countries—from detection by fruit flies.

The findings of Atilano, Pereira et al. highlight how bacteria employ a number of ways to evade detection. If similar behavior is observed when bacteria infect humans, autolysins could represent a new drug target for the treatment of bacterial diseases.

---

level of PGN detection. This is achieved through specific PGRPs: PGRP-LC specifically recognizes DAP-type PGN, usually found in Gram-negative bacteria and Gram-positive bacilli, and activates the Imd pathway (**Leulier et al., 2003**), while PGRP-SA recognizes lysine-type PGN, which surrounds most Gram-positive bacteria, and activates the Toll pathway (**Lemaitre and Hoffmann, 2007**). Activation of either pathway results in a series of multiple defense reactions that include the production and secretion of antimicrobial peptides into the hemolymph of flies (**Lemaitre and Hoffmann, 2007**). In mammals, PGRPs can act as antibacterial agents due to their bactericidal and/or bacteriostatic activity, mediated by PGN hydrolytic activity (e.g. PGLYRP-2 [**Dziarski and Gupta, 2006**]) or by the binding of PGRPs to targets on the bacterial cell surface, which causes the activation of specific bacterial two-component systems, resulting in bacteria killing through a mechanism that includes membrane depolarization and production of hydroxyl radicals (**Kashyap et al., 2011**).

Bacterial PGN is concealed by an outer membrane in Gram-negative bacteria, or by layers of proteins and glycopolymers in Gram-positive bacteria. It is therefore usually assumed that an infected organism only recognizes PGN in the form of fragments released into the surrounding medium by the activity of different bacterial or host enzymes (**Nigro et al., 2008**). However, it has been recently shown that PGRP-SA can directly bind PGN at the bacterial surface in conditions such as the absence of wall teichoic acids (WTAs) (**Atilano et al., 2011**). Therefore, it is possible that bacteria may have developed different strategies to prevent host receptors from binding PGN on the bacterial surface, thus avoiding detection by the host innate immune system.

In order to look for molecules with a role in preventing bacterial recognition by the host, we used PGRP-SA, a host receptor circulating in the hemolymph of *Drosophila melanogaster*, to test if specific proteins involved in the metabolism of bacterial cell wall were required to conceal the PGN at the surface of the Gram-positive pathogenic bacterium *Staphylococcus aureus*. We have identified the

major autolysin Atl as an essential protein for concealing the bacterial PGN at the cell surface from host detection.

## Results

### PGN hydrolytic enzymatic activities encoded by the *atl* gene are required to conceal *S. aureus* from Drosophila PGN receptor PGRP-SA

In order to identify factors that Gram-positive bacteria use to conceal their PGN present at the bacterial cell surface from host recognition, we have constructed *S. aureus* null mutants lacking non-essential genes involved in PGN metabolism and determined the ability of a fluorescent derivative of PGRP-SA (mCherry_PGRP-SA) to bind to their surfaces (*Figure 1*). Specifically, we tested mutants constructed in NCTC8325-4 *S. aureus* strain, expressing altered levels of autolysins (Δ*arlR*), lacking the major autolysin (Δ*atl*) or factors that modulate autolysin activity (Δ*fmtA*), producing non-O-acetylated peptidoglycan with increased susceptibility to lysozyme (Δ*oat*), or producing altered WTAs lacking attached β-O-GlcNAc (Δ*tarS*) or D-alanyl residues (Δ*dltA*). A mutant lacking the major autolysin Atl, a PGN hydrolase, was identified as the most severely impaired in its ability to avoid recognition by mCherry_PGRP-SA.

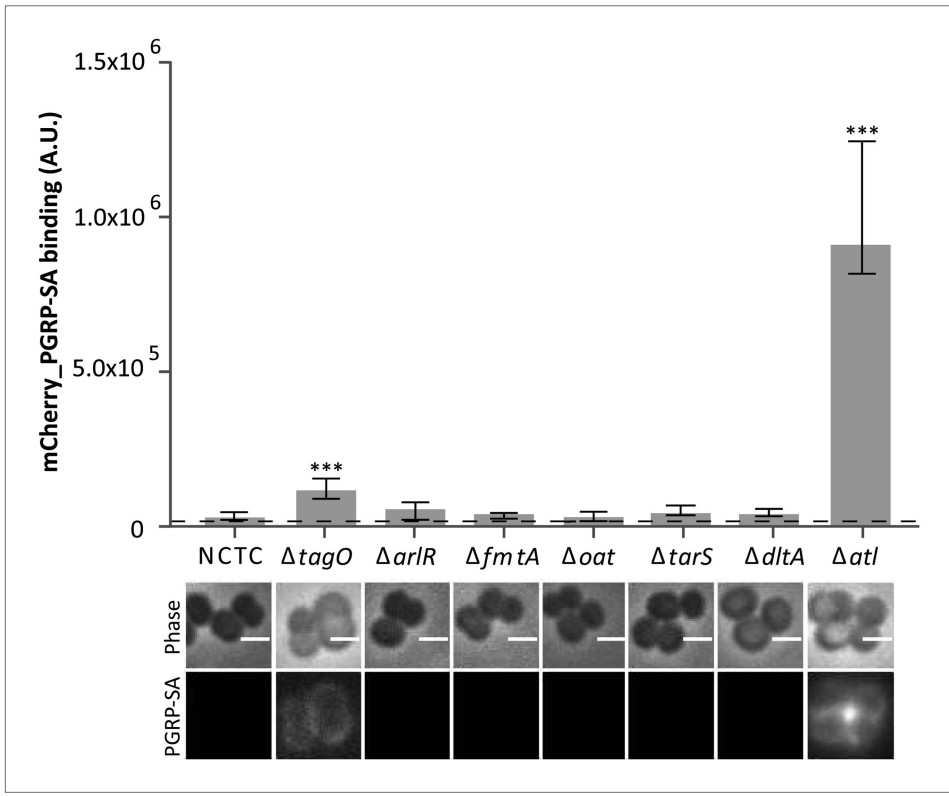

**Figure 1**. *Staphylococcus aureus* cells lacking the major autolysin Atl are better recognized by mCherry_PGRP-SA. Exponentially growing bacteria from the parental NCTC8325-4 (NCTC), and its mutant strains lacking genes involved in cell wall metabolism (see main text for details) were incubated with mCherry_PGRP-SA in 96-well plates. Cells were pelleted by centrifugation and unbound protein was washed with PBS. mCherry_PGRP-SA bound to each bacterial strain was quantified using a fluorescent image analyzer (n≥ 10 wells for each strain). Results are shown as the median with 25% and 75% inter-quartile range. The dashed line represents the median value obtained when bacteria were absent. Statistically significant differences (p<0.001, indicated by asterisks) were observed only between mCherry_PGRP-SA binding to the parental strain and mutants NCTCΔ*tagO* and NCTCΔ*atl*. mCherry_PGRP-SA binding to bacterial cells was also imaged by fluorescence microscopy. The top panels show phase-contrast images of bacteria (white scale bar represents 1 μm) and the bottom panels show the mCherry_PGRP-SA binding to their surface. Binding of the protein was observed for NCTCΔ*tagO* and NCTCΔ*atl*, with the later exhibiting the highest binding.

PGN modifications, namely the attachment of teichoic acids or the degree of PGN polymerization (defined as the ratio between the amount of polymerized muropeptides and the amount of monomeric muropeptides present in the PGN macromolecule), have been previously associated with changes in the recognition of *S. aureus* PGN by host receptors (*Filipe et al., 2005*; *Atilano et al., 2011*). However, an *S. aureus atl* null mutant produces a PGN with a muropeptide composition similar to the parental strain (*Figure 2A*), that is, it shows no increase in the amount of polymerized muropeptides, which were previously reported to be better inducers of an innate immune response than monomeric muropeptides (*Filipe et al., 2005*). Moreover, the *S. aureus atl* null mutant produces WTAs (*Figure 2B*), indicating that the mechanism to conceal PGN dependent on the presence of *atl* is new, and different from that previously reported for a WTA mutant (*Atilano et al., 2011*). In agreement, the simultaneous deletion of *tagO* and *atl* had a synergistic effect on exposing the surface PGN to host recognition (*Figure 2C*).

The *atl* gene encodes a polypeptide that is cleaved into two proteins capable of digesting PGN: an amidase (AM), which cleaves the amide bond between peptides and glycans, and a glucosaminidase (GL), which cleaves glycan strands (*Figure 3A*). To determine which one of these activities was required for the role of Atl in PGN concealment, we constructed *S. aureus* strains expressing, from the *atl* native chromosomal locus, Atl mutants with impaired amidase activity (AM$^{H265A}$ [*Bose et al., 2012*]), glucosaminidase activity (GL$^{E1128A}$ [*Bose et al., 2012*]), or both (Atl$^{H265A/E1128A}$ double mutant) and confirmed the absence of the expected hydrolytic activity in a zymogram (*Figure 3B*).

When the *atl* mutants were incubated with mCherry-PGRP-SA, both the mutant having only glucosaminidase activity (AM$^{H265A}$ mutant) and the mutant having only amidase activity (GL$^{E1128A}$ mutant) were still able to avoid strong PGRP-SA binding, indicating that either amidase or glucosaminidase enzymatic activity was sufficient to impair recognition of *S. aureus* by PGRP-SA (*Figure 3C*). Only when both activities were absent (Atl$^{H265A/E1128A}$ double mutant) was mCherry-PGRP-SA capable of easily recognizing the bacterial cell surface (*Figure 3C,D*), albeit to lower levels than those observed for the *atl* null mutant. This difference may be due to residual amidase or glucosaminidase activity in the *S. aureus* Atl$^{H265A/E1128A}$ double mutant, not detectable in a zymogram.

The proposed role for the amidase and glucosaminidase activities in concealing bacteria was confirmed using purified enzymes. Pre-incubation of *S. aureus atl* null mutant cells with either AM or GL (approximately 0.4 μM) completely abolished mCherry_PGRP-SA binding to the bacterial surface (*Figure 4A*). When both enzymes were added simultaneously to the *atl* null mutant cells, lower concentrations of each enzyme (<0.02 μM) were sufficient to conceal bacteria from the host immune receptor (*Figure 4B*). This is in agreement with the synergy in PGRP-SA binding observed for the *S. aureus* Atl$^{H265A/E1128}$ double mutant (*Figure 2C*).

We then questioned if the mechanism used by autolysins to conceal PGN at the bacterial surface was dependent on their lytic activity. An alternative hypothesis could be that amidases, or other proteins capable of binding PGN, prevented binding of PGRP-SA by competing for the same PGN substrate, given that the structural fold of PGRPs closely resembles the fold of amidases (*Zoll et al., 2010*). To distinguish between the two hypotheses, we first incubated *S. aureus atl* null mutant cells with purified AM or with GL, and then washed them thoroughly to ensure complete removal of the proteins (*Figure 5*). This procedure did not increase recognition by PGRP-SA, showing that the activity of the autolysins, and not their physical presence, was required for protection.

## Trimming of the bacterial cell surface by Atl PGN hydrolytic activities restores the ability of *S. aureus atl* mutant bacteria to kill infected *Drosophila*

The results described above show that amidase or glucosaminidase activities can efficiently protect *S. aureus* cells from direct recognition by the *Drosophila* PGN receptor PGRP-SA. Does this protection confer an advantage to *S. aureus* cells during infection in the *Drosophila* model? Indeed, Tabuchi et al. have previously shown that *S. aureus atl* mutants are impaired in their virulence (*Tabuchi et al., 2010*). We have confirmed that the *atl* mutant used in this work, when compared to the parental *S. aureus* strain NCTC8325-4, was also severely impaired in its ability to kill wild type *Drosophila*, both the 25174 strain, recently established from wild caught flies (*Mackay et al., 2012*), as well as the yw strain, which is the parental lineage used to generate the PGRP *seml* mutant (*Michel et al., 2001*; *Figure 6A,B*). Furthermore, the *atl* mutant triggered a stronger induction of drosomycin expression (frequently used as a read-out for Toll activity) when normalized for the same number of infecting bacteria (*Figure 6C,D*).

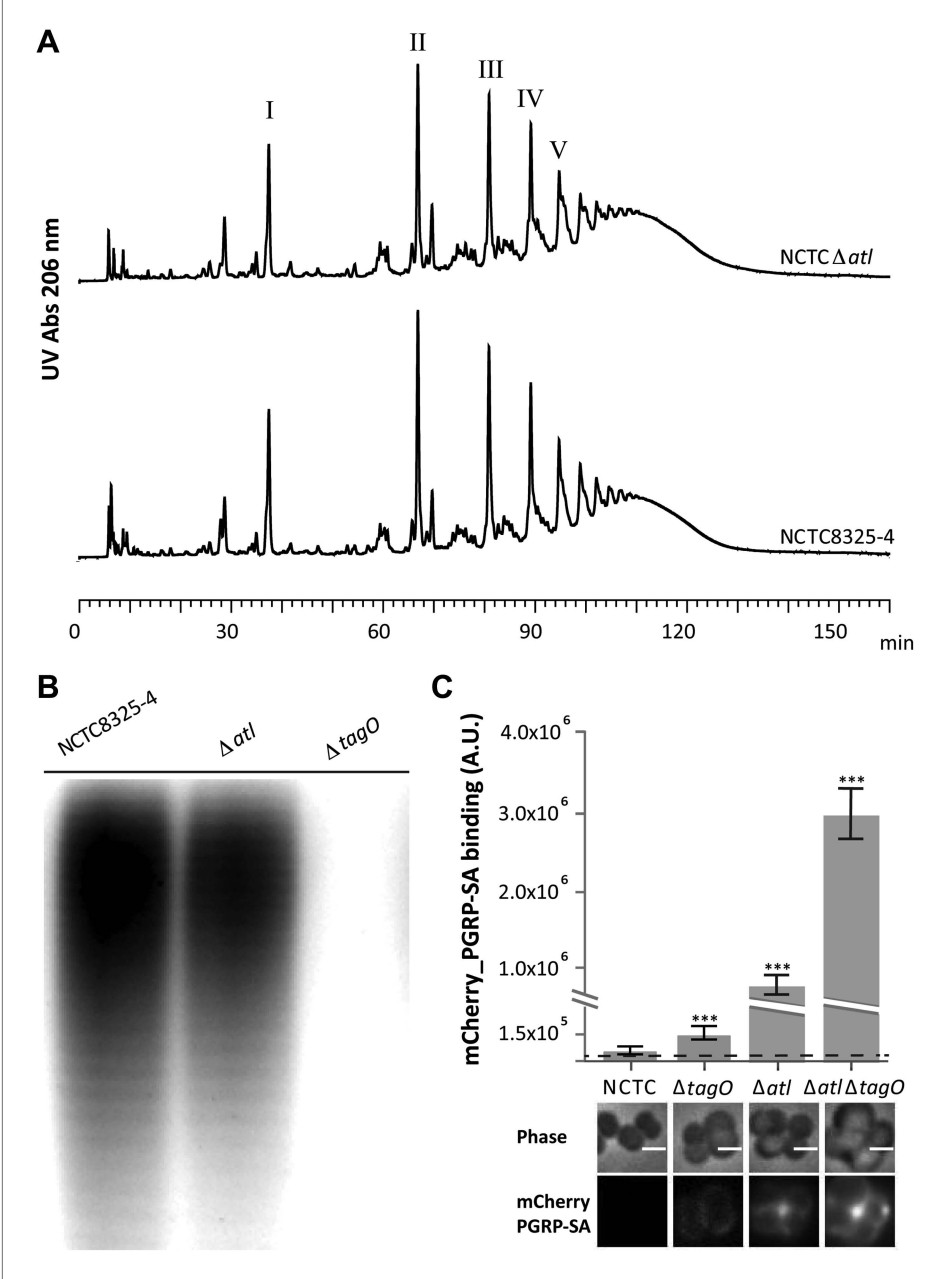

**Figure 2**. Better recognition of *Staphylococcus aureus* NCTC*Δatl* by PGRP-SA is not mediated by alterations in peptidoglycan muropeptide composition or lack of wall teichoic acids production. (**A**) *Staphylococcus aureus* NCTC*Δatl* mutant has a similar peptidoglycan (PGN) muropeptide composition to the parental strain NCTC8325-4, as seen by HPLC analysis of mutanolysin-digested PGN. Roman numerals I to V indicate muropeptide species form monomers to pentamers, respectively. (**B**) NCTC*Δatl* mutant produces wall teichoic acids (WTAs), as shown by PAGE analysis of surface WTAs from NCTC8325-4, NCTC*Δatl*, and NCTC*ΔtagO* (which lacks WTAs). (**C**) Deletion of both the *tagO* and *atl* genes had a synergistic effect on PGN exposure to the host receptor PGRP-SA, indicating independent mechanisms of WTA and Atl in PGN concealment. *S. aureus* parental strain NCTC8325-4 and mutant strains NCTC*ΔtagO*, NCTC*Δatl*, and NCTC*ΔatlΔtagO* were incubated with mCherry_PGRP-SA in 96-well plates. The average amount of mCherry_PGRP-SA bound to bacteria in each well was quantified using a fluorescent image analyzer (n = 10 wells, for each strain), and is represented as the median with 25% and 75% inter-quartile range. The dashed line represents the median value obtained with control samples (no bacteria added). Statistically significant differences (p<0.001) are indicated by asterisks and were observed between mCherry_PGRP-SA binding to the parental strain and each mutant as well as between mutants. mCherry_PGRP-SA binding to the cells was also

*Figure 2. Continued on next page*

*Figure 2. Continued*

confirmed by fluorescence microscopy (bottom). Gray panels show phase-contrast images of bacterial cells (white scale bar represents 1 μm) and black panels show mCherry_PGRP-SA binding.

Importantly, our data showed that survival of *Drosophila* infected with an *S. aureus atl* mutant was dependent upon the presence of a functional PGRP-SA (*Figure 6E*). Furthermore, these mutant bacteria were capable of propagating similarly to the parental *S. aureus* strain in flies that lacked a functional PGRP-SA (*Figure 6F*), showing that impaired virulence is not due to a lower proliferation rate of the *atl* mutant.

Two, non-mutually exclusive, hypotheses can explain the decreased virulence of the *atl* mutant. (1) *S. aureus* wild type bacteria produce amidases that cleave the stem peptide moiety of soluble PGN fragments. Given that stem peptides are required for binding of PGRPs to PGN, amidase activity would therefore reduce the inflammatory activity of the soluble PGN. In contrast, the *atl* mutant would release intact muropeptides, easily detected by PGRP-SA, and therefore would be unable to evade detection by the innate immune system. (2) Amidases produced by *S. aureus* wild type bacteria would shave and remove the most accessible PGN fragments at the surface of bacteria, eliminating the binding sites for PGRPs. As the *atl* mutant does not produce the Atl amidase, it would have extending PGN fragments at its surface, which would be easily detected by PGRP-SA, inducing an immune response.

If the first hypothesis is correct, inactivation of the amidase activity, but not of the glucosaminidase (which cleaves the glycans and therefore releases intact, inflammatory PGN fragments), should result in decreased virulence. Alternatively, if the second hypothesis is mainly correct, inactivation of either the amidase or the glucosaminidase activity should result in decreased virulence, as both activities can remove fragments of detectable PGN (containing peptides) from the bacterial surface. We therefore tested the virulence of bacteria with either impaired amidase activity (AM$^{H265A}$) or impaired glucosaminidase activity (GL$^{E1128A}$) and observed that both had a reduced ability to kill flies (*Figure 7A*) in accordance with the second hypothesis. Moreover, injection of *seml Drosophila* with *atl* mutant bacteria pre-coated with mCherry_PGRP-SA resulted in increased fly survival (*Figure 7B*), again favoring a role for PGRP-SA in the recognition of PGN directly at the bacterial surface. Nevertheless, the amount of PGRP-SA present at the surface of the injected bacteria may not have been sufficient to result in full complementation of the ability of *seml* flies to survive infection. This is in accordance with the hypothesis that a continuous and high expression of PGRP-SA during the entire infection process is required to ensure survival of the infected host (*De Gregorio et al., 2001*).

## Amidase and glucosaminidase activities trim the bacterial cell surface of neighboring bacteria and prevent bacterial recognition by Drosophila PGN receptor PGRP-SA

Secreted autolysins not only bind to bacterial cell surfaces but they can also be found in the growth medium (*Pasztor et al., 2010*). This raises an interesting hypothesis: can secreted bacterial autolysins protect neighboring bacteria? If this was true, autolysin producers could in theory protect an entire bacterial population from recognition by the host. To test this hypothesis we collected and filtered the supernatant from a culture of *S. aureus* parental strain NCTC8325-4 (Atl producer) and incubated *S. aureus atl* null mutant cells in this medium for 30 min. The *atl*-encoded proteins secreted into the supernatant by the parental strain were capable of modifying the surface of the *atl* null mutant cells, consequently abolishing their recognition by PGRP-SA (*Figure 8A,B*). This was not observed when supernatant from a culture of *S. aureus atl* null mutant was used (*Figure 8A,B*), showing that *atl* products (and not other secreted PGN hydrolases) are responsible for PGN concealment.

Importantly, we observed that *atl* mutants regained the ability to kill flies if pre-treated with a supernatant from a culture of the Atl producer NCTC8325-4 but not if pre-treated with supernatant from a culture of the *atl* mutant (*Figure 8C*).

Moreover, previously bound PGRP-SA could be removed by autolysin activity present in the medium, as seen by time-lapse microscopy of *S. aureus atl* null mutant cells initially covered by bound mCherry_PGRP-SA and then incubated with supernatant containing amidase and glucosaminidase enzymes (*Figure 9*).

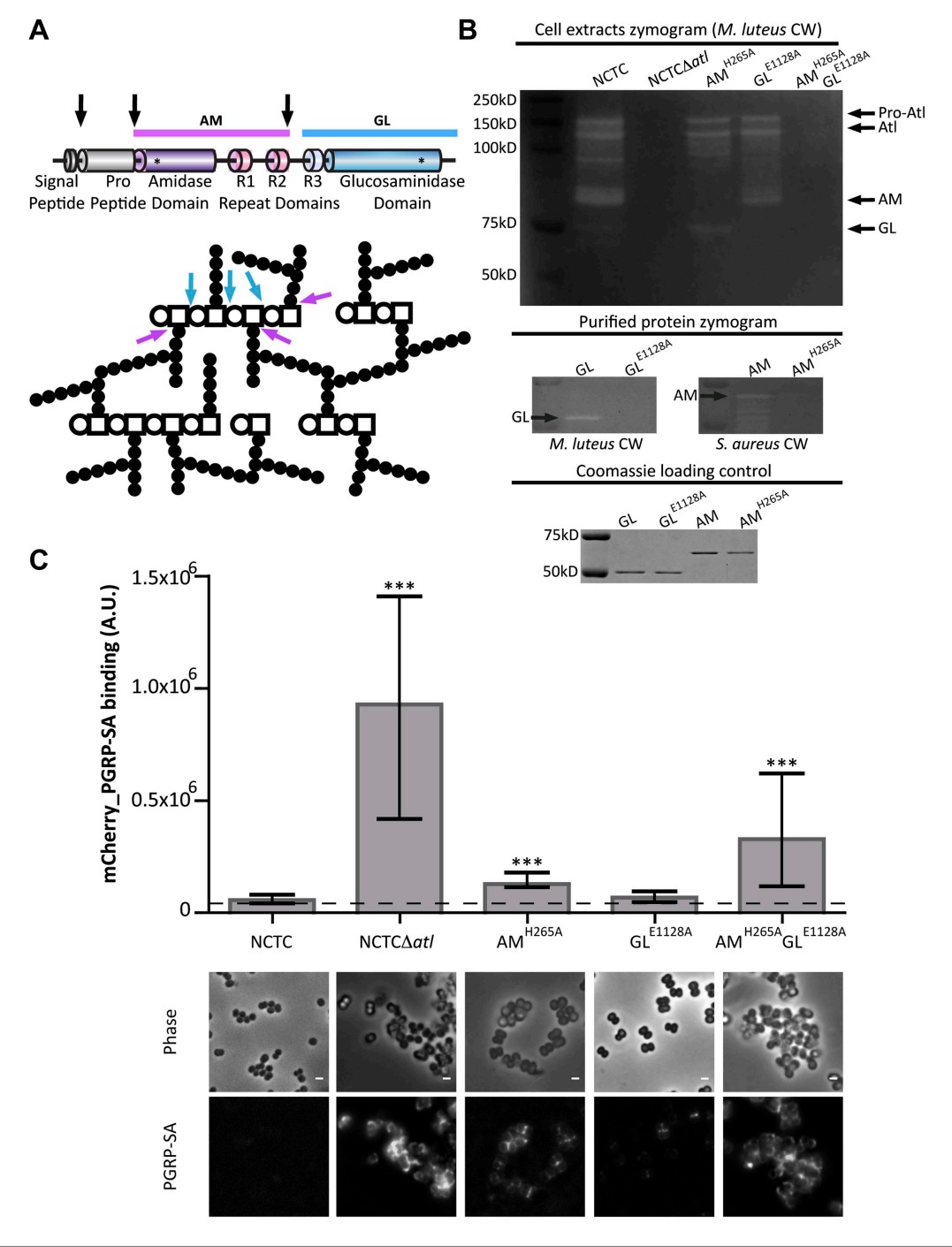

**Figure 3**. Amidase and glucosaminidase activities limit mCherry_PGRP-SA binding to the bacterial cell surface. (**A**) Representation of the post-translational cleavage (black arrows) of the *atl* encoded protein. The processed amidase (AM, purple arrow) releases peptidoglycan (PGN) stem peptides by cutting the bond between the stem peptides (black circles) and the N-acetylmuramic acid residue (open square). The glucosaminidase (GL, blue arrow) releases muropeptides by cutting the glycosidic linkage between N-acetylmuramic acid (open square) and N-acetylglucosamine (open circle). For simplicity, only some examples of AM and GL cleavage sites are indicated by arrows. (**B**) Zymogram analysis of the activity of autolysins extracted from different *Staphylococcus aureus atl* mutant strains expressing inactive amidase (AM^H265A), inactive glucosaminidase (GL^E1128A), or both (AM^H265A GL^E1128A). Top: Cells were harvested at mid-exponential phase, and protein extracts were prepared and
*Figure 3. Continued on next page*

*Figure 3. Continued*

run in an SDS–PAGE gel containing *Micrococcus luteus* cell walls. PGN hydrolytic activity is seen as clear bands in the stained gel and showed that *S. aureus* encoding mutant proteins AM or GM were lacking only the expected activity. Middle: A similar analysis was carried out with purified proteins AM, GL, and their inactive forms, AM[H265A] and GL[E1128A], respectively, showing that mutant proteins were not active. Bottom: SDS–PAGE of the protein samples loaded into the zymograms confirmed that similar amount of proteins were used. (**C**) *S. aureus* parental and *atl* mutant strains were incubated for mCherry_PGRP-SA in 96-well plates. mCherry_PGRP-SA bound to each bacterial strain is represented as the median with 25% and 75% inter-quartile range (n = 50 wells). A significant increase of mCherry_PGRP-SA binding, relative to the parental strain, was observed with *S. aureus* mutants expressing AM[H265A] and AM[H265A] GL[E1128A] (p<0.0001), but not the GL[E1128A] mutant. mCherry_PGRP-SA binding to *S. aureus* parental and *atl* mutant strains was also confirmed by fluorescence microscopy. Gray panels are phase-contrast images of bacterial cells (white scale bar represents 1 μm) and black panels show mCherry_PGRP-SA binding.

Together, these data suggest that the autolysins encoded by *atl* can protect not only the producer cells but also a population of invading bacteria from detection by the immune system during infection, enabling it to kill the infected host.

### Role of autolysins in evasion from host PGN receptor is conserved in highly virulent CA-MRSA *S. aureus* and in *Streptococcus pneumoniae*

The *S. aureus* strain used in this work, NCTC8325-4, is susceptible to most antibiotics and has limited interest from a clinical point of view. We therefore tested if the role of *atl*-encoded enzymes in making PGN inaccessible to the host was conserved in other strains, namely in methicillin-resistant *S. aureus* (MRSA) strains. MRSA strains are a leading cause of bacterial infections in hospitals and an important cause of community-acquired (CA) bacterial infections in the United States. MW2 is a particularly virulent MRSA strain that caused the earliest reported cases of CA-MRSA infection in the United States (*Chambers and Deleo, 2009*). Deletion of the *atl* gene in MW2 led to higher binding of mCherry_PGRP-SA, that is, to improved recognition of the bacterial surface (*Figure 10A*). More importantly, MW2Δ*atl* cells were severely impaired in virulence, as demonstrated by the decreased capacity to kill *Drosophila* (*Figure 10B*).

The fact that PGN sensing is an evolutionarily conserved recognition mechanism in innate immunity, suggests that the strategy used by *S. aureus* to conceal PGN from the host could be effective across different animal phyla and even used by other bacteria. We therefore tested if we could observe this phenomenon in another Gram-positive pathogenic bacteria, namely *Streptococcus pneumoniae*, a frequent cause of pneumonia in developed countries and a major cause of infant mortality by septicemia in developing countries (*Kadioglu et al., 2008*). Lack of LytA, the major amidase of *S. pneumoniae*, resulted in bacteria that were better recognized by mCherry_PGRP-SA, a phenotype that was reversed when LytA was expressed from a plasmid (*Figure 11A*). Interestingly, it is possible that autolysins can participate in inter-species protection, as *S. pneumoniae* LytA, an amidase that releases PGN stem peptides, and *Streptomyces globisporus* mutanolysin, a commercially available muramidase that releases muropeptides, were capable of protecting *S. aureus atl* null mutant cells from PGRP-SA recognition (*Figure 11B*).

These results suggest a general role for autolysins, such as amidases, in protecting bacteria from host recognition.

## Discussion

We propose that bacteria have evolved various independent mechanisms to prevent direct recognition by PGN host receptors. We have previously shown that the presence of glycopolymers, such as WTAs, at the bacterial surface contributes to *S. aureus* evasion from the *Drosophila* innate immune system (*Atilano et al., 2011*). We now show that Atl also protects bacteria from host recognition but by a different mechanism, given that simultaneous deletion of *tagO* (essential for teichoic acid biosynthesis) and *atl* had a synergistic effect on exposing the surface PGN to host recognition, when compared to individual deletion of each of these genes, as shown in *Figure 2C*. This new role for Atl is in accordance with previous reports that have shown in the *Drosophila* infection model that *S. aureus atl* mutants are impaired in their virulence (*Tabuchi et al., 2010*).

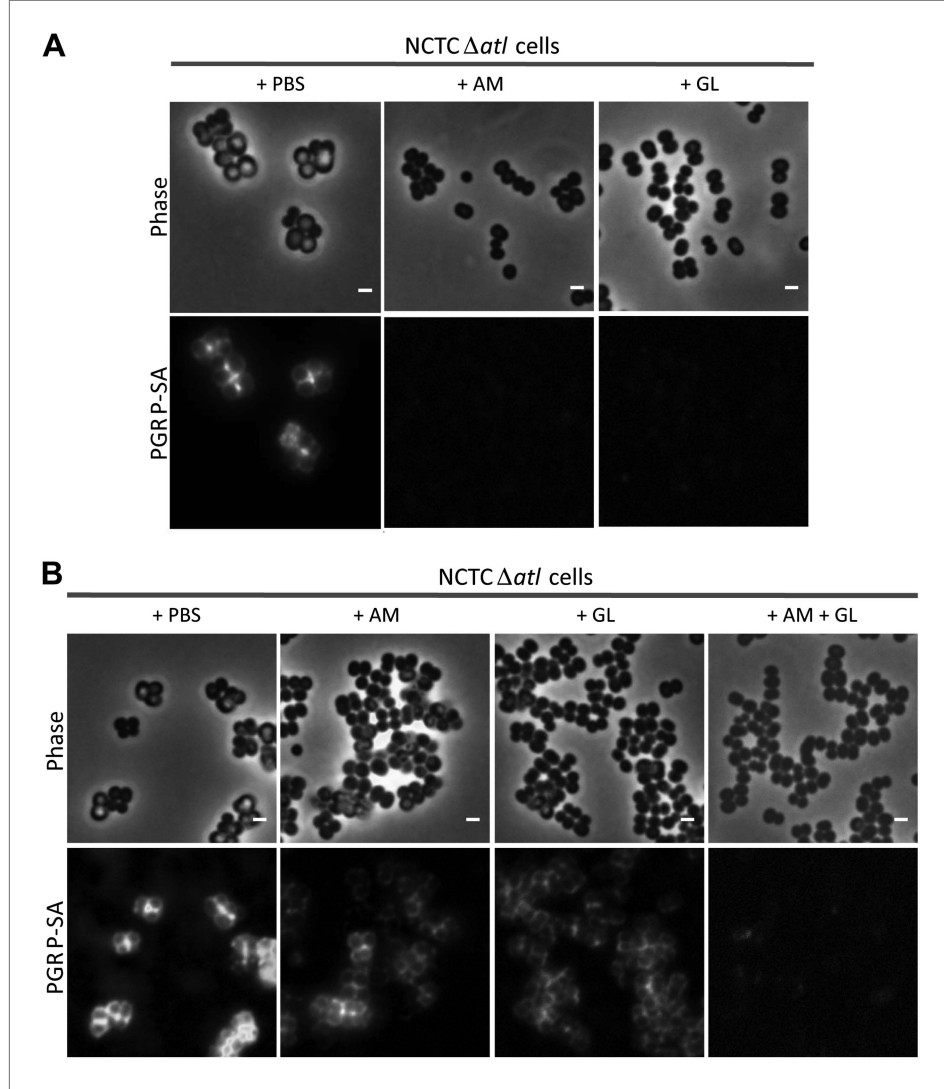

**Figure 4**. Combination of purified staphylococcal AM and GL completely abolishes binding of mCherry_PGRP-SA to the surface *Staphylococcus aureus atl* null mutant cells. (**A**) mCherry_PGRP-SA binding to NCTCΔ*atl* bacteria pre-incubated with purified AM and GL (0.37 μM and 0.44 μM, respectively) showed that both enzymes can impair binding of mCherry_PGRP-SA to the cell surface. PBS was used as negative control. (**B**) The use of lower concentrations of purified AM and GL in combination, to treat NCTCΔ*atl Staphylococcus aureus* cells (final concentrations of 7.9 nM and 18.7 nM, respectively) was more effective than the single use of AM (final concentration of 7.9 nM) or GL (final concentration of 18.7 nM) in preventing binding of mCherry-PGRP-SA to the surface of NCTCΔ*atl*. Gray panels are phase-contrast images of bacterial cells (white scale bar represents 1 μm) and black panels show mCherry_PGRP-SA binding.

Bacterial autolysins, or murein hydrolases, are present in most bacteria and have been recognized as important players in bacterial metabolism. Autolysins are required, among other roles, to cleave the pre-existing PGN structure so that new PGN subunits are incorporated during synthesis, to ensure daughter cell separation after complete bacterial division or to prompt antibiotic-induced lysis of bacterial cells (*Vollmer et al., 2008*; *Uehara and Bernhardt, 2011*). Autolysins have also been recognized as important virulence factors, as they may act as adhesins (*Hell et al., 1998*) or be required for biofilm formation (*Vollmer et al., 2008*; *Uehara and Bernhardt, 2011*).

The new role for Atl uncovered in this work was surprising, since it has been previously suggested that PGN fragments, released by host enzymes capable of degrading PGN or shed by dividing bacteria during the course of infection, may facilitate host recognition in the context of both NOD receptors and insect

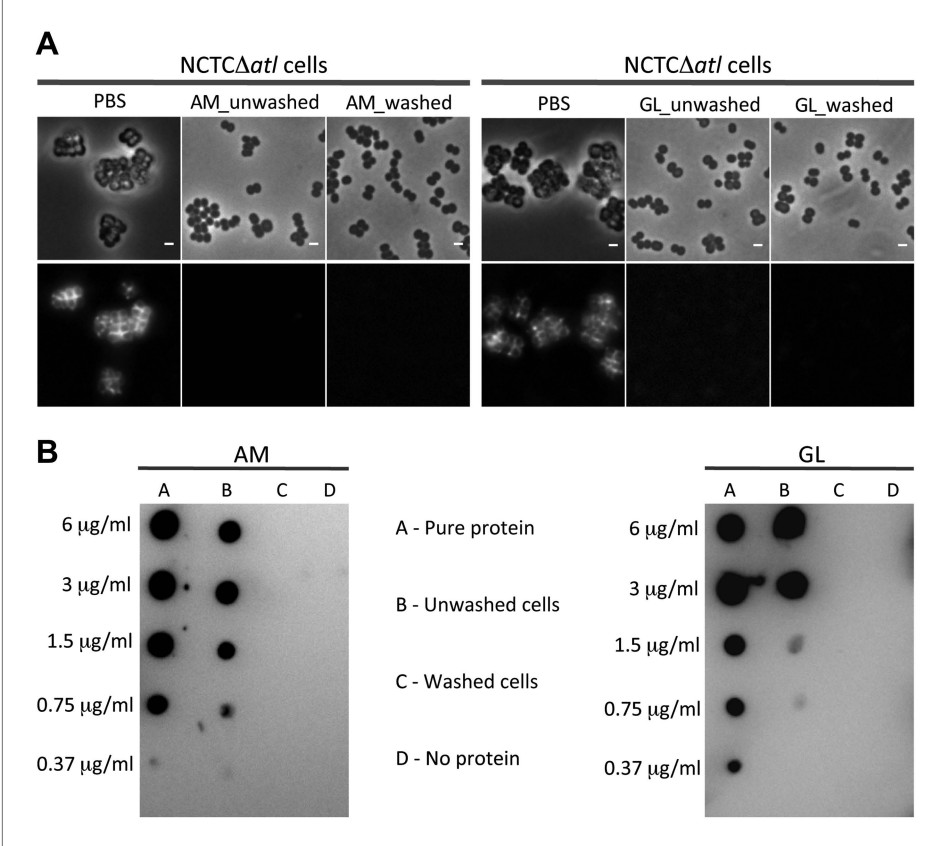

**Figure 5**. Activity, and not the presence of the Atl products, is required to avoid bacterial surface recognition by mCherry_PGRP-SA. (**A**) NCTCΔ*atl* cells were incubated with purified staphylococcal AM, GL, or PBS (negative control) and were either washed to clear these proteins from the cell surface, or unwashed to keep the proteins attached. Bacterial cells were then labeled with mCherry_PGRP-SA and imaged by fluorescence microscopy. In both cases (washed and unwashed), mCherry_PGRP-SA was unable to bind *S. aureus* NCTCΔ*atl* cells, showing that the physical presence of the enzyme is not required for this effect. Gray panels are phase-contrast images of bacterial cells (white scale bar corresponds to 1 μm) and black panels are fluorescence microscopy images showing binding of mCherry_PGRP-SA to the surface of bacteria. (**B**) To confirm that AM (left) and GL (right) were absent from the washed samples, a dot-blot assay using anti-His antibody, which recognizes the His-tagged AM and GL enzymes, was performed before the addition of mCherry_PGRP-SA. Washed cells (lanes **C**) showed no presence of protein, whereas in unwashed cells (lanes **B**) the presence of each individual lytic enzyme was detected (detection limit for AM and GL is lower than 0.37 μg/ml, as seen in lanes **A**, which corresponds to the pure protein loaded onto the membrane at different concentrations). A sample of cells to which no protein was added was used as a negative control (lanes **D**).

PGRPs (*Nigro et al., 2008*). However, we have now shown that it is precisely this autolysin-mediated 'trimming' of PGN (which presumably releases PGN fragments) that renders *S. aureus* inaccessible to the host.

Direct recognition of PGN seems less likely to play a role in mammalian cells, where PGN is recognized by the intracellular pattern-recognition NOD factors, or in the recognition of Gram-negative bacteria, whose PGN is concealed by the outer membrane, by the *Drosophila* innate immune system.

We propose that a major mechanism of Gram-positive bacteria recognition in innate immunity is the direct binding of PGRPs to PGN on whole bacteria, which have dedicated proteins, namely amidases or glucosamidases, to trim exposed fragments of PGN. These fragments may extend beyond the teichoic acids layer and, through their removal, bacteria avoid their own recognition and possibly that of their neighbors, by the host innate immune system. Therefore, targeting bacterial autolysins in order to prevent their activity may constitute a new approach to decrease bacterial virulence and potentiate the effectiveness of the host immune system.

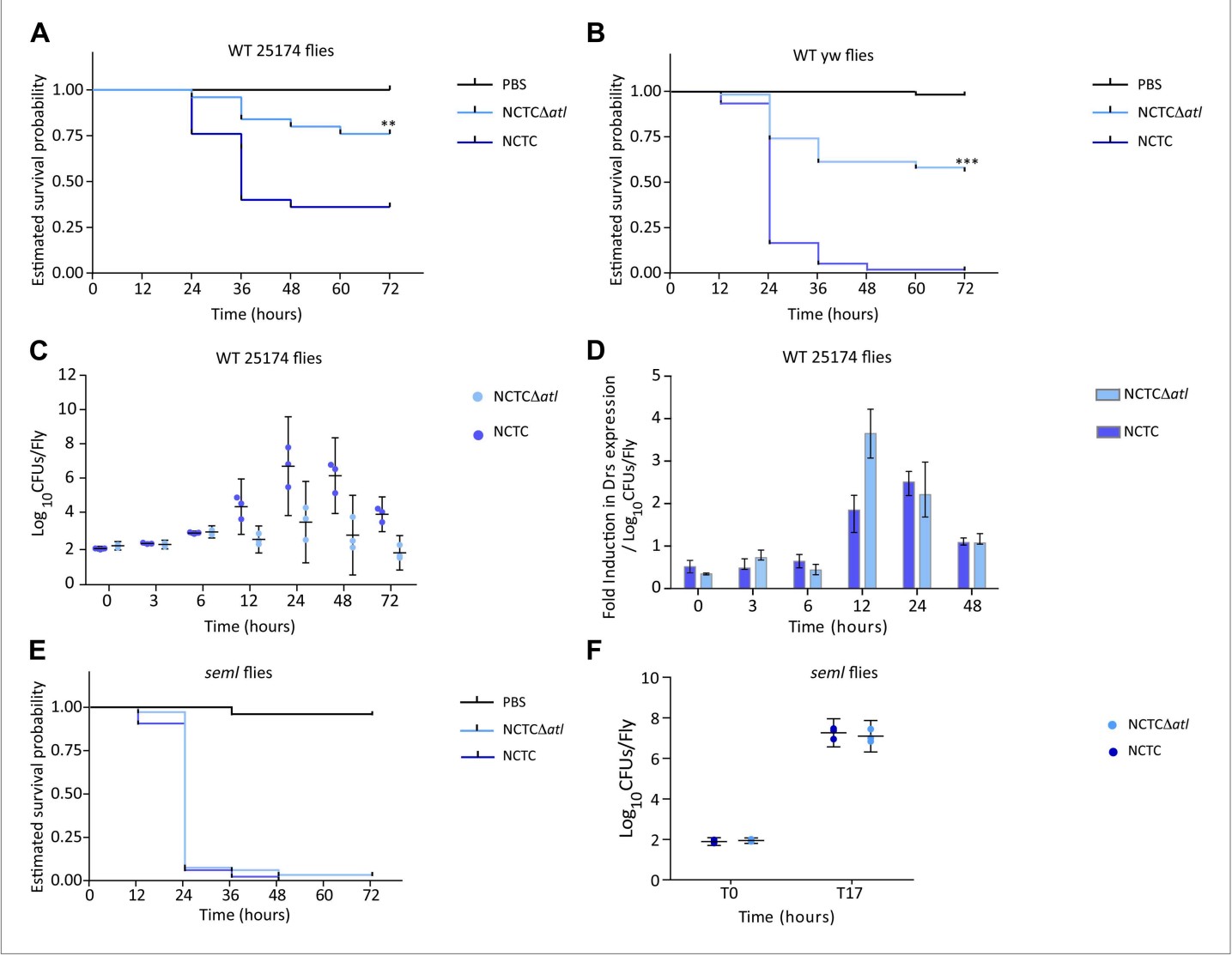

**Figure 6**. PGRP-SA is required to control infection by a *Staphylococcus aureus* mutant lacking the major autolysin Atl. (**A**) Estimated survival curves of wild type (WT) 25174 flies infected with Atl producer NCTC8325-4 and NCTCΔ*atl* strains were statistically different (p<0.005) and are indicated by asterisks. Absence of Atl activity in the *S. aureus* Δ*atl* mutant resulted in bacteria that have a decreased ability to kill WT flies. Survival of infected flies (n = 75) was monitored at 12 h intervals for 3 days. (**B**) Estimated survival curves of WT yw flies, the parental lineage used to construct the peptidoglycan recognition protein (PGRP) *semI* mutant used in this study, were also produced as described above and gave similar results (p<0.0001, indicated by asterisks). (**C**) The number of bacteria harvested from infected flies at different time points after infection, during survival assays, was determined by plating in *S. aureus* growth medium and counting CFUs. (**D**) Drosomycin (Drs) expression was determined by qPCR at different time points after infection and is shown after normalization for the number of infecting bacteria present in flies (panel **C**). A stronger induction of drosomycin expression was observed in flies injected with NCTCΔ*atl* bacteria, 12 h after infection. (**E**) PGRP-SA mutant flies succumbed equally well to infection by NCTC832-4 and NCTCΔ*atl* bacteria (p>0.05), showing that in the absence of a functional PGRP-SA, NCTCΔ*atl* bacteria can proliferate and kill the infected host. (**F**) Similar numbers of NCTC832-4 and NCTCΔ*atl* bacteria were present in PGRP-SA mutant flies 17 h after infection (T17), confirming that the *atl* mutant proliferated as well as the parental bacterial strain in *semI* flies.

# Materials and methods

## Bacterial strains and growth conditions

All bacterial strains used in this study are listed in **Supplementary file 1**. *Escherichia coli* strains were grown in Luria–Bertani broth (LB; Difco, France) or Luria–Bertani agar (LA; Difco) medium at 37°C with aeration. When needed, the antibiotics ampicillin (Amp; Sigma-Aldrich, Germany) and erythromycin

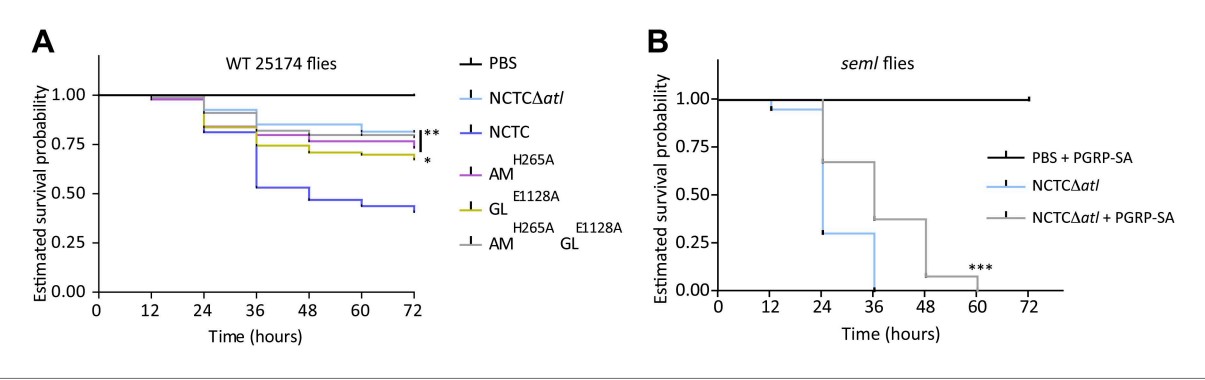

**Figure 7**. Lack of either Atl amidase or glucosaminidase activity, which leads to different types of released peptidoglycan fragments, results in decreased virulence of *Staphylococcus aureus*. (**A**) Estimated survival curves of wild type (WT) 25174 flies infected with *atl* mutant strains impaired in amidase activity (AM$^{H265A}$), glucosaminidase activity (GL$^{E1128A}$), or both (AM$^{H265A}$GL$^{E1128A}$), were statistically different (p<0.05) from survival curves of flies infected with the Atl producer NCTC8325-4 strain, but not with the NCTCΔ*atl* strain. Statistically significant differences (*p=0.01 and **p<0.001) are indicated by asterisks and were observed between the estimated survival curve of flies infected with the parental bacteria strain NCTC8325-4 and each survival curve of flies infected with the different *atl* mutant strains. WT 25174 flies infected with the different *atl* mutant strains succumbed in a similar manner (p>0.05). (**B**) Estimated survival curve of PGRP-SA *seml* mutant flies infected with NCTCΔ*atl* bacteria pre-coated with mCherry-PGRP-SA showed that direct binding of the peptidoglycan (PGN) host receptor to the surface resulted in an increase of resistance of *seml* mutant flies to bacterial infection. Statistically significant differences (p<0.0001) are indicated by asterisks and were observed between the two estimated survival curves.

(Ery; Sigma-Aldrich) were added at a final concentration of 100 µg/ml, and kanamycin (Km; Sigma-Aldrich) was added at a final concentration of 30 µg/ml. *Lactococcus lactis* LL108 strain was grown in M17 broth (Difco), supplemented with sucrose (0.5 M) and glucose (0.5% wt/vol) at 30°C without aeration. Erythromycin was used when required at 100 µg/ml. *S. aureus* strains were grown at 30°C with aeration in tryptic soy broth (TSB; Difco) or on tryptic soy agar (TSA; Difco). Medium was supplemented when required with Ery at 10 µg/ml and/or 5-bromo-4-chloro-3-indolyl β-D-galactopyranoside (X-Gal; Apollo Scientific, UK) at 100 µg/ml. *S. pneumoniae* was grown in C + Y medium at 37°C, without aeration, or on TSA plates supplemented with 5% vol/vol sheep blood (Probiológica, Portugal). Tetracycline (Sigma-Aldrich) and Ery were used, when required, at 1 µg/ml and 0.25 µg/ml, respectively.

## Fly strains
Isogenic *Drosophila* Bloomington #25174 and #6599 (yw) were used as wild type flies. PGRP_SA$^{seml}$ flies were used as a PGRP-SA mutant background (*Michel et al., 2001*). Stocks were raised on standard cornmeal-agar medium at 25°C.

## Construction of *S. aureus* and *S. pneumoniae* mutants
All plasmids used in this study are listed in *Supplementary file 1* and the sequences of the primers used are listed in *Supplementary file 2*.

The *S. aureus oatA* and *fmtA* null mutants were constructed using the integrative vector pORI280 (*Leenhouts et al., 1996*). To delete the *oatA* gene from the genome of *S. aureus* NCTC8325-4 we amplified two 1.3 Kb DNA fragments from NCTC8325-4 chromosomal DNA, corresponding to the upstream (primers P1_oatA and P2_oatA) and downstream (primers P3_oatA and P4_oatA) regions of the *oatA* gene. The upstream fragment was digested with *Bam*HI and *Nco*I and cloned into pORI280. The downstream PCR fragment was digested with *Nco*I and *Bgl*II and then cloned into the previous plasmid, which already contained the upstream fragment, producing the plasmid pΔ*oatA*. To delete the *fmtA* gene, we amplified two DNA fragments of approximately 0.9 Kb from *S. aureus* NCTC8325-4 chromosomal DNA, corresponding to the upstream (primers P5_fmtA and P6_fmtA) and downstream (primers P7_fmtA and P8_fmtA) regions of the *fmtA* gene. These fragments were joined by overlap PCR using primers P5_fmtA and P8_fmtA. The resulting PCR product was digested with *Bam*HI and *Bgl*II, and cloned in the pORI280 plasmid, producing the plasmid pΔ*fmtA*. Plasmids pΔ*oatA* and pΔ*fmtA* were sequenced and electroporated into the transformable RN4220 strain at 30°C (using Ery selection) as previously described (*Veiga and Pinho, 2009*). The plasmids were then transduced to

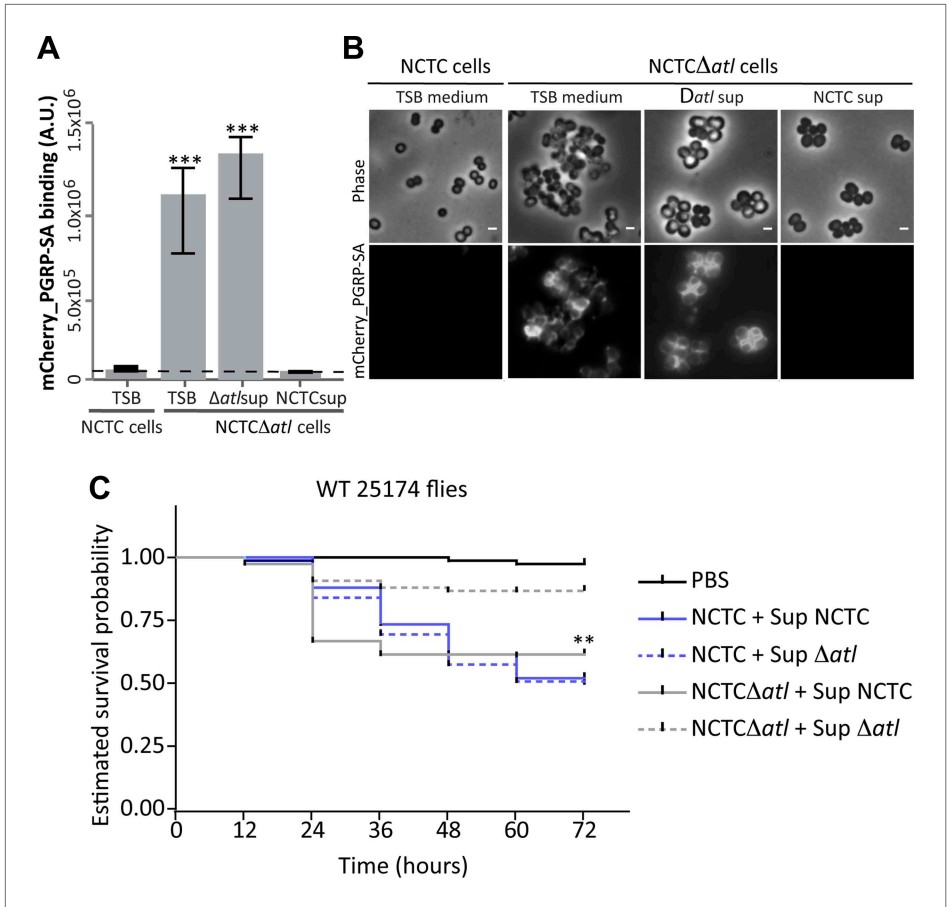

**Figure 8**. Atl products secreted by *Staphylococcus aureus* cells protect *atl* null mutant cells from PGRP-SA recognition and allow *S. aureus* to establish a successful infection in Drosophila. (**A**) NCTCΔ*atl Staphylococcus aureus* cells were incubated with TSB medium (control) or with supernatants (sterilized by filtration) from cultures of NCTC8325-4 (NCTCsup, containing *atl* encoded products) or NCTCΔ*atl* (Δ*atl*sup) strains. After washing with PBS, cells were mixed with mCherry_PGRP-SA in 96-well plates. Binding of the protein to the cells was determined as described in *Figure 1*. mCherry_PGRP-SA binding to NCTCΔ*atl* cells pre-incubated with supernatant from Atl producer strain NCTC8325-4 was 100-fold lower than to the same cells pre-incubated with supernatant from a culture of NCTCΔ*atl* mutant. The dashed line represents the median value obtained with no bacteria. Statistically significant differences (p<0.001) are indicated by asterisks. (**B**) Similar results were observed by fluorescence microscopy of NCTCΔ*atl S. aureus* cells incubated with filtered supernatants from NCTC8325-4 or NCTCΔ*atl* cells (TSB medium was used as negative control) and subsequently labeled with mCherry_PGRP-SA. Only the supernatant from Atl producer NCTC8325-4 modified the surface of NCTCΔ*atl* cells and limited mCherry_PGRP-SA binding. The top panels are phase-contrast images of bacterial cells (white scale bar represents 1 μm) and the bottom panels show the mCherry_PGRP-SA binding to the bacterial surface. (**C**) Estimated survival curves for wild type Drosophila infected with *S. aureus* NCTC8325-4 and NCTCΔ*atl* strains that were pre-incubated with bacteria-free supernatant from overnight cultures of each of the *S. aureus* strains. Flies were infected with approximately 100 *S. aureus* CFUs and fly survival was assessed every 12 h over 3 days. As expected, survival curves for flies infected with NCTC8325-4 bacteria pre-incubated with both supernatants were statistically indistinguishable (p>0.05). Treatment of NCTCΔ*atl* cells with both supernatants resulted in distinct survival curves (p<0.001, indicated by asterisks), showing that NCTCΔ*atl* cells recover the ability to kill flies if pre-incubated with supernatant from a culture of the Atl producer NCTC8325-4 strain.

NCTC8325-4 using the phage 80α. The resulting strains, carrying the plasmids integrated into the chromosome, were grown in liquid media, without antibiotic and at 37°C, for several generations to allow for loss of the plasmid and, consequently, of the *lacZ* and *erm* genes. Different dilutions of the cultures were plated on TSA plates containing X-Gal (without Ery) and incubated at 37°C overnight. White colonies were isolated and their sensitivity to erythromycin was confirmed. Absence of the *oatA*

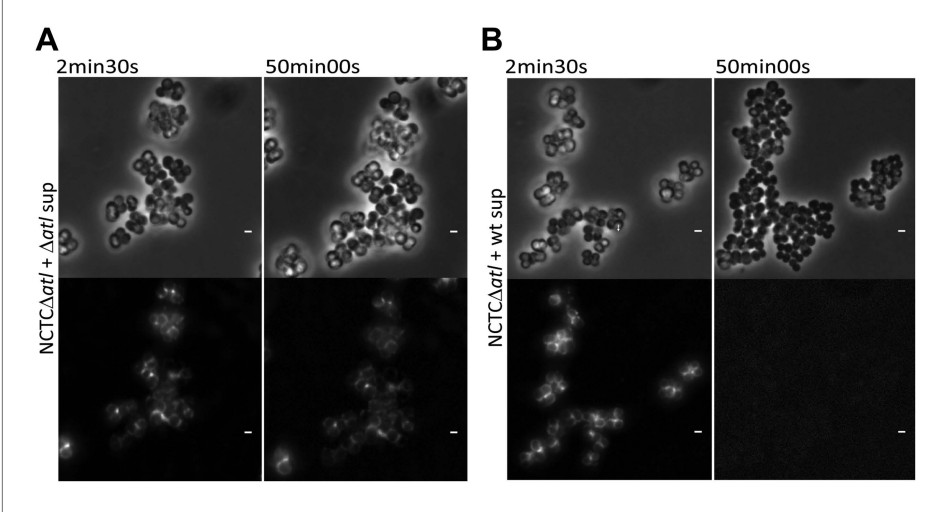

**Figure 9**. Time-lapse microscopy showing that *atl* encoded proteins can mediate release of mCherry_PGRP-SA previously bound to the surface of *atl* null mutant *Staphylococcus aureus* cells. NCTCΔ*atl* cells, labeled with mCherry-PGRP-SA, were placed on top of a thin layer of agarose containing filter-sterilized supernatant of cultures of NCTCΔ*atl* mutant (**A**, *Video 1*) or NCTC8325-4 parental strain (**B**, *Video 2*), and observed by fluorescence microscopy in a time-lapse experiment. The supernatant from the Atl producer parental strain (**B**), in contrast to the supernatant from the *atl* null mutant (**A**), triggered the release of mCherry-PGRP-SA previously attached to the bacterial cell surface. Gray panels are phase-contrast images of bacterial cells (white scale bar corresponds to 1 µm) and black panels are fluorescence microscopy images showing binding of mCherry_PGRP-SA to the surface of bacteria.

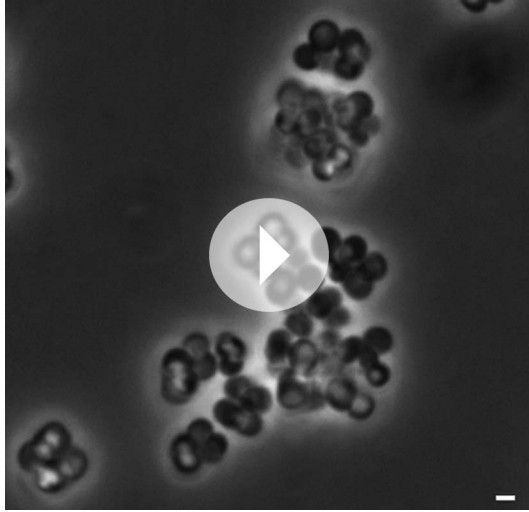

**Video 1**.

and *fmtA* genes (NCTCΔ*oatA* and NCTCΔ*fmtA*) was shown by PCR and confirmed by sequence analysis of the amplified DNA fragment.

All other *S. aureus* null mutants described were constructed using the thermosensitive vector pMAD (*Arnaud et al., 2004*). In order to generate the *arlR* null mutant, 0.8 Kb PCR fragments of the upstream and downstream regions of the *arlR* gene were amplified from chromosomal DNA of NCTC8325-4, using primer pairs P9_arlR/P10_arlR and P11_arlR/P12_arlR. The two PCR products were joined in an overlap PCR reaction using the primers P9_arlR/P12_arlR. The resulting product was digested with *Nco*I and *Bgl*II and cloned into pMAD, giving rise to the pΔ*arlR* plasmid.

To delete the *atl* gene, the pΔ*atl* plasmid was constructed by amplifying 1 Kb fragments from the downstream and upstream regions of the *atl* gene, using the primer pairs P13_atl/P14_atl and P15_atl/P16_atl. These two fragments were joined by overlap PCR using the primers P13_atl and P16_atl, and the obtained PCR product was digested with *Bgl*II and *Eco*RI and cloned into the pMAD vector, giving rise to the pΔ*atl* plasmid.

To delete the *tarS* gene, we amplified by PCR two DNA fragments of approximately 0.6 Kb, corresponding to the upstream (primers P17_tarS/P18_tarS) and downstream (primers P19_tarS/P20_tarS) regions of the *tarS* gene. The two fragments were joined by overlap PCR, using primers P17_tarS and P20_tarS. The resulting PCR product was digested with *Bgl*II and *Nco*I and cloned into pMAD vector, producing the plasmid pΔ*tarS*.

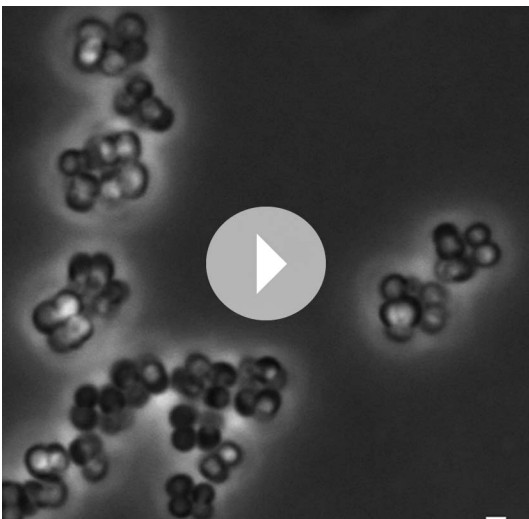

**Video 2**.

To delete the *dltA* gene, two 0.55 Kb DNA fragments were amplified by PCR from the genome of *S. aureus* NCTC8325-4, corresponding to the upstream (primers P25_dltA/P26_dltA) and downstream (primers P27_dltA/P28_dltA) regions of the *dltA* gene. The two fragments were joined by overlap PCR using primers P25_dltA and P28_dltA, and the resulting PCR product was digested with *Bgl*II and *Eco*RI and cloned into the pMAD vector, producing the plasmid pΔ*dltA*.

The pΔ*arlR*, pΔ*atl*, pΔ*tarS*, and pΔ*dltA* plasmids were sequenced and electroporated into *S. aureus* RN4220 strain at 30°C (using Ery selection) and then transduced to NCTC8325-4 using phage 80α. Insertion and excision of the plasmids for deletion of the genes (*arlR*, *atl*, *tarS*, and *dltA*) from the NCTC8325-4 chromosome was completed as previously described (*Arnaud et al., 2004*), with the exception of the NCTCΔ*dltA* strain where the last step was performed at 30°C due to the thermosensitive nature of cells lacking the *dltA* gene. All gene deletions were confirmed by PCR and the resulting strains were named NCTCΔ*arlR*, NCTCΔ*atl*, NCTCΔ*tarS*, and NCTCΔ*dltA*.

To make a NCTCΔ*atl*Δ*tagO* double mutant, the pΔ*tagO* plasmid (*Atilano et al., 2010*) was transduced into the NCTCΔ*atl* strain using phage 80α, at 30°C due to the thermosensitive nature of cells lacking the *tagO* gene. The double mutant was identified among the white colonies sensitive to Ery, by PCR and DNA sequencing, as described above.

The point mutants in the Atl amidase (AM) and Atl glucosaminidase (GL) domains were constructed by site-directed mutagenesis. To inactivate the AM domain, the conserved amino acid H265 was exchanged for an alanine (H265A mutation). To inactivate the GL domain, the amino acid residue E1128 was mutated to an alanine (E1128A mutation). To generate the H265A mutation, a 794 bp region upstream from the codon encoding H265 (primers P29_AtlAM and P30_AtlAM) and a 1070 bp region downstream from the codon encoding H265 (primers P31_AtlAM and P32_AtlAM) were amplified from NCTC8325-4 genomic DNA by PCR. Joining of the up and downstream regions by overlap PCR (primers P29_AtlAM and P32_AtlAM) resulted in the amplification of the DNA fragment encoding the mutated Atl amidase domain, which was cloned into the pMAD vector, using *Bam*HI and *Eco*RI restriction enzymes, to construct plasmid pAM[H265A]. A similar approach was used to generate the Atl point mutation in the Atl glucosaminidase (GL) domain. Briefly, a 659 bp fragment upstream of the codon encoding E1128 (primers P33_AtlGL and P34_AtlGL) and a 680 bp fragment downstream of the codon encoding E1128 (primers P35_AtlGL and P36_AtlGL) were amplified by PCR. The two fragments were joined by overlap PCR using primers P33_AtlGL and P36_AtlGL, digested and cloned into the *Eco*RI/*Bam*HI sites of the pMAD vector, yielding plasmid pGL[E1128A]. After sequencing, plasmids pAM[H265A] and pGL[E1128A] were electroporated into RN4220 and then transduced into NCTC8325-4. Insertion into and excision from the chromosome generated *S. aureus* point mutants in the amidase and glucosaminidase domains. The strains carrying the Atl mutations were verified by sequencing and were named AM[H265A] and GL[E1128A]. A double mutant, named Atl[H265A/E1128A], mutated in both the amidase and glucosaminidase domains, was constructed by phage transduction of plasmid pGL[1128] into the AM[H265A] strain, followed by insertion into and excision from the chromosome, as described above.

The *S. pneumoniae lytA* null mutant strain was constructed in the background of R36A strain, using the integrative vector pORI280. To delete the *lytA* gene, we amplified two 1.0 Kb DNA fragments from R36A chromosomal DNA, corresponding to the upstream (primers P41_LytA and P42_LytA) and downstream (primers P43_LytA and P44_LytA) regions of the *lytA* gene. The two fragments were then joined by overlap PCR using primers P41_LytA and P44_LytA. The PCR product was digested with *Bam*HI and *Eco*RI and cloned into plasmid pORI280, producing the plasmid pΔ*lytA*. This plasmid was propagated in *L. lactis* and purified before being transformed into the unencapsulated *S. pneumoniae* R36A strain. The R36AΔ*lytA* strain was obtained by insertion and excision of the plasmid into the chromosome, as

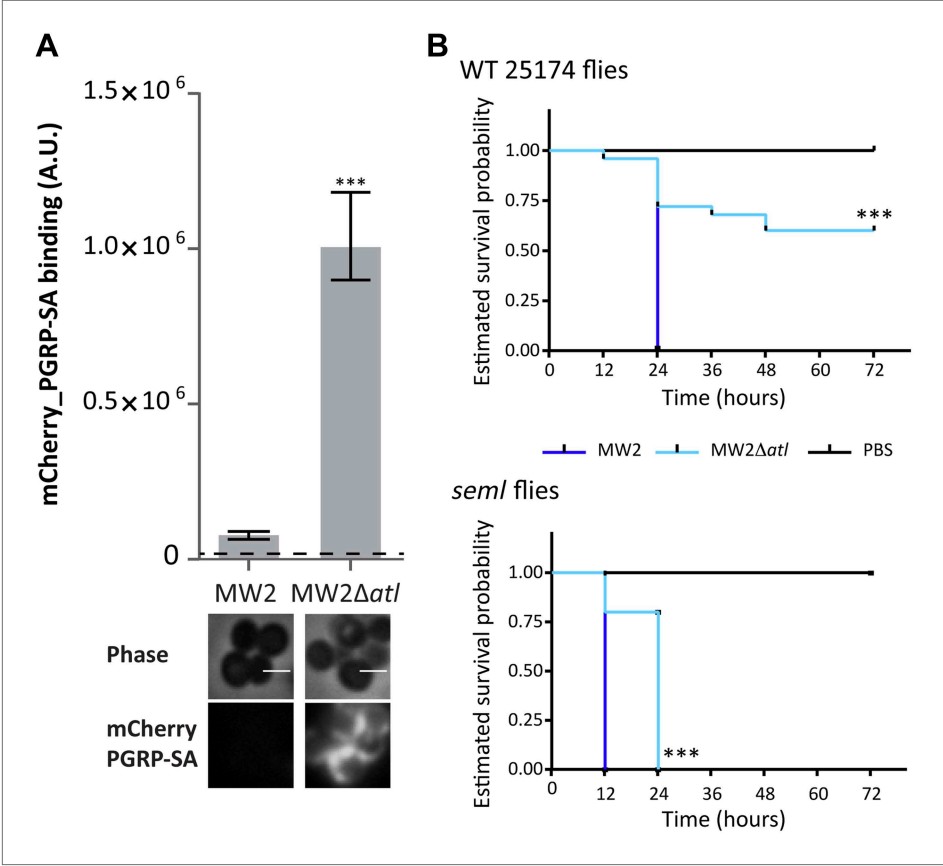

**Figure 10**. Atl encoded proteins are required to conceal CA-MRSA virulent strains from the Drosophila immune system. (**A**) *Staphylococcus aureus* CA-MRSA MW2 strain and its *atl* null mutant, MW2Δ*atl*, were incubated with mCherry_PGRP-SA in 96-well plates. The average amount of mCherry_PGRP-SA bound to bacteria in each well was quantified using a fluorescent image analyzer (n = 10 wells, for each strain), and is represented as the median with 25% and 75% inter-quartile range. The dashed line represents the median value obtained with control sample (no bacteria added). mCherry_PGRP-SA binding to MW2 cells was significantly different ($p<0.05$, indicated by asterisks) from the binding to MW2Δ*atl*. mCherry_PGRP-SA binding to the bacterial cell surface of MW2 and MW2Δ*atl* bacteria was also confirmed by fluorescence microscopy (bottom). Gray panels are phase-contrast images of bacterial cells (white scale bar corresponds to 1 μm) and black panels show mCherry_PGRP-SA binding. (**B**) Estimated survival curves for wild type (WT) and PGRP-SA deficient (*semi*) flies infected with *S. aureus* MW2 and MW2Δ*atl*. MW2Δ*atl* is impaired in its ability to kill WT flies, showing that lack of Atl strongly reduces MW2 pathogenicity. PGRP-SA deficient flies are killed in less than 24 h by both *S. aureus* strains, showing that flies control MW2Δ*atl* infection in a PGRP-SA dependent manner. Statistically significant differences ($p<0.0001$) are indicated by asterisks and were observed between the two estimated survival curves.

previously described (**Leenhouts et al., 1996**). For the construction of a plasmid expressing LytA (p*lytA*), the *lytA* gene from *S. pneumoniae* R36A genome was amplified with primers P45_LytA and P46_LytA, digested with *Nhe*I and *Bgl*II, and cloned into plasmid pBCSLF001 (**Henriques et al., 2011**).

## Construction of recombinant plasmids for AM and GL protein expression

A 3177 bp fragment of the *atl* gene, missing the pro-peptide sequence, was amplified by PCR using genomic DNA of *S. aureus* strain COL with primers Pexp1 and Pexp4, which included *Bam*HI and *Sal*I restriction sites. The DNA fragment was purified, digested with the respective restriction enzymes, and cloned into pET28a(+) plasmid (Novagen, Merck Millipore, UK) using *E. coli* DH5α, yielding pET-AMR$_1$R$_2$R$_3$GL.

To construct a plasmid expressing the AM recombinant protein, plasmid pET-AMR$_1$R$_2$R$_3$GL was used as the template for site-directed mutagenesis using primers Pexp_stop1 and Pexp_stop2, to change

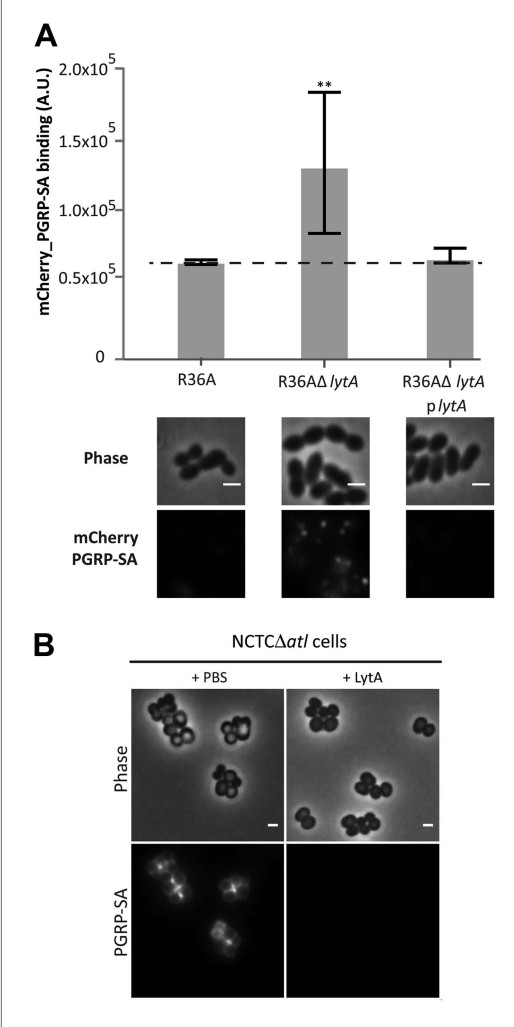

**Figure 11**. LytA activity can prevent *Streptococcus pneumoniae* and *Staphylococcus aureus* recognition by mCherry_PGRP-SA. (**A**) *Streptococcus pneumoniae* parental strain R36A, a *lytA* null mutant (R36AΔ*lytA*), and a complemented strain, expressing LytA from a replicative plasmid (R36AΔ*lytA*p*lytA*), were incubated with mCherry_PGRP-SA in 96-well plates. The average amount of mCherry_PGRP-SA bound to bacteria in each well was quantified using a fluorescent image analyzer (n = 10 wells, for each strain), and is represented as the median with 25% and 75% inter-quartile range. The dashed line represents the median value obtained with control sample (no bacteria added). mCherry_PGRP-SA binding to R36AΔ*lytA* was significantly higher (p<0.05, indicated by asterisks) than that observed for the parental strain or R36AΔ*lytA*p*lytA* strain, showing that deletion of the *lytA* gene in *S. pneumoniae* increases surface recognition by mCherry_PGRP-SA. Binding of mCherry_PGRP-SA to *S. pneumoniae* cells was also imaged using fluorescence microscopy (bottom). Gray panels are phase-contrast images of bacterial cells (white scale bar represents 1 µm) and black panels show the mCherry_PGRP-SA binding to *Figure 11. Continued on next page*

codon 775 from AAA (Lys) to TAA (stop codon) resulting in pET-AMR$_1$R$_2$.

To construct a plasmid expressing the GL recombinant protein, primers Pexp2 and Pexp4 with *Bam*HI and *Sal*I restriction sites, were used to amplify a 1446 bp sequence encoding for the GL domain of Atl, using COL genomic DNA as the template. The DNA fragment was purified, digested with the respective restriction enzymes, and cloned into pET28a(+) plasmid using *E. coli* DH5α, yielding pET-R$_3$GL.

To construct a plasmid expressing protein AM$^{H265A}$, pET-AMR$_1$R$_2$ was used as the template for site-directed mutagenesis using primers Pexp_H265A1 and Pexp_H265A2, to change codon 265 from GTA (His) to GCA (Ala). A plasmid expressing protein GL$^{E1128A}$ was constructed similarly, using pET-R$_3$GL as the template for directed mutagenesis with primers Pexp_E1228A1 and Pexp_E1228A2, changing codon GAA (Glu) to GCA (Ala). The amplified plasmids were digested with *Dpn*I prior to transformation into *E. coli* DH5α, resulting in pET-AM$^{H265A}$ and pET-GL$^{E1228A}$.

The recombinant plasmids (*Supplementary file 1*) were confirmed by restriction analysis and sequencing and used to transform *E. coli* BL21(DE3) strain for protein expression.

## Protein purification

A functional mCherry-tagged derivative of PGRP-SA protein, mCherry-PGRP-SA, was expressed in *E. coli* and purified using a cobalt affinity resin (Clontech) as previously described (*Atilano et al., 2011*).

The AM and GL proteins were expressed as N-terminal 6×His-tag fusion proteins. Expression of the recombinant proteins was performed using an auto-induction based expression method. Cells were grown for 18 h at 37°C with shaking, harvested, and resuspended in 1/10 vol of purification lysis buffer (50 mM NaH$_2$PO$_4$, 10 mM imidazole, 300 mM NaCl, pH 8.0) containing 10 U/ml of benzonase nuclease (Novagen, Merck Millipore) and Complete-Mini Protease Inhibitor Cocktail (Roche, Switzerland). After cell disruption, the lysates were cleared. The recombinant proteins were soluble and therefore present in the supernatant.

Protein purification was performed using Ni-NTA agarose columns (Qiagen, Germany) under native conditions, according to the manufacturer's instructions. The recombinant proteins were eluted with purification elution buffer (50 mM NaH$_2$PO$_4$, 250 mM imidazole, 300 mM NaCl, pH 8.0). The expression and purification yields were monitored by SDS–PAGE. The most concentrated elution fractions were dialyzed for 16 h in a 3500 MWCO

*Figure 11. Continued*

bacteria. (**B**) *S. aureus* NCTCΔ*atl* cells were incubated with purified pneumococcal amidase (LytA) or with the commercially available mutanolysin, a muramidase from *Streptomyces globisporus* capable of degrading peptidoglycan into its muropeptide components. Bacterial cells were washed to clear these proteins from the cell surface, labeled with mCherry_PGRP-SA, and imaged by fluorescence microscopy, which showed that both pneumococcal LytA amidase and mutanolysin can prevent detection of NCTCΔ*atl* S. *aureus* cells by mCherry_PGRP-SA.

SnakeSkin (Pierce Biotechnology, Rockford, IL) at 4°C, against 100 mM Tris, pH 7.5.

For the purification of *S. pneumoniae* LytA amidase, *E. coli* BL21(DE3) cells were transformed with the plasmid pGL100 (*Garcia et al., 1987*) and incubated overnight at 37°C, with vigorous shaking, in LB supplemented with 100 μg/ml ampicillin and 2% lactose. Cells were harvested by centrifugation, resuspended in 20 mM sodium phosphate buffer, pH 6.9, and broken by sonication. Clarified lysate was applied to DEAE-cellulose resin and incubated at 4°C for 1 h with stirring. Bound protein was washed five times with 20 mM sodium phosphate buffer containing 1.5 M NaCl, then eluted in the same buffer containing 2% choline. Protein was dialyzed against 20 mM sodium phosphate buffer, pH 6.9 to remove the choline and aliquots were stored at −20°C.

## mCherry_PGRP-SA binding to bacteria

To quantify mCherry_PGRP-SA binding to bacterial cells, cultures were grown to mid-exponential phase ($OD_{600\ nm}$ 0.5), centrifuged, and resuspended in TSB to an OD of 2.5.

Approximately 100 μl of each culture was placed into wells of a 96-well plate and harvested at room temperature (RT) for 5 min at 3600×*g*. Cells were washed and then resuspended in 100 μl of sterile PBS (pH 6). mCherry_PGRP-SA was added to each well at a final concentration of 60 μg/ml, followed by incubation for 5 min at RT. Cells were then harvested and washed with PBS. The fluorescent signal of the mCherry_PGRP-SA was detected using a 532 nm laser in a Fuji FLA-5100 reader (FUJIFILM Life Science) and the fluorescence intensity of each well was determined using ImageJ software (*Abràmoff et al., 2004*).

To image the mCherry_PGRP-SA binding to bacterial cell surface by fluorescence microscopy, cells were treated as previously described (*Atilano et al., 2011*), except that mCherry_PGRP-SA was added to a final concentration of ≈300 μg/ml, in a reaction volume of 200 μl. After labeling, bacterial cells were centrifuged, washed twice with 200 μl of PBS, resuspended in 20 μl of PBS, and observed by fluorescence microscopy.

## Preparation of crude autolytic enzyme extracts and zymographic analysis

Preparation and analysis of crude autolytic enzyme extracts from *Staphylococcus aureus* is described in detail at Bio-protocol (*Vaz and Filipe, 2015*). Crude autolytic enzyme extracts were prepared from *S. aureus* cultures grown in TSB to $OD_{600}$ 0.3 at 30°C. The cells were harvested by centrifugation (7000 rpm, 15 min at 4°C), and the cell pellet was washed in 15 ml of ice-cold 50 mM Tris–HCl (pH 7.5), 150 mM NaCl. Cells were resuspended in SDS 4% (500 ml) and incubated for 30 min at 25°C with stirring. Zymographic analysis of the proteins' peptidoglycan hydrolytic activities was performed as described earlier (*Heilmann et al., 1997*). Briefly, crude cell lysates or pure proteins isolated as described above were separated by SDS–PAGE on a polyacrylamide gel (10% acrylamide/0.2% bisacrylamide) containing inactivated *Micrococcus luteus* or *S. aureus* cells (recovered at mid-exponential phase, washed with water, and heat inactivated by autoclaving at 121°C for 15 min) as a substrate in the resolving gel. Equal amounts of crude extracts (50 μg) or purified proteins (200 ng), quantified in a Nanodrop ND-1000 spectrophotometer (Thermo Scientific, Wilmington, NC), were loaded in the gel. After electrophoresis, gels were washed in deionized H2O for 15 min at RT, and then incubated in renaturation buffer (0.1% Triton X-100, 10 mM CaCl2, 10 mM MgCl2, 50 mM Tris–HCl, pH 7.5) at 37°C with gentle agitation. The zymogram was stained in methylene blue solution (1% methylene blue in KOH 0.01%) for 3 min and destained in water until bands with peptidoglycan hydrolytic activity were observed as clear zones in the opaque gel.

### WTA analysis

WTAs were extracted by alkaline hydrolysis from overnight cultures, analyzed by native PAGE, and visualized by combined alcian blue/silver staining, as previously described (*Atilano et al., 2010*).

### Peptidoglycan purification and analysis

Mutanolysin (Sigma)-digested peptidoglycan was prepared from NCTC8325-4 and NCTCΔ*atl S. aureus* strains as previously described (*Filipe et al., 2005*). The released muropeptides were reduced with sodium borohydride and separated by HPLC on a $C_{18}$ column (ODS-Hypersyl, 5 μm, 4.6×250 mm;

Thermo Scientific) at a flow rate of 0.5 ml/min for 160 min with a gradient from 5% to 30% (vol/vol) methanol in 100 mM $NaH_2PO_4$, pH 2.0. The eluted muropeptides were detected by their UV absorption at 206 nm, and their abundance was estimated by calculating the percentage of the integrated area of each peak relative to the total area of the muropeptide peaks, using Shimadzu LC Solution software.

## Effect of secreted Atl products on PGRP-SA binding to the surface of bacterial cells

Cell-free supernatants from the *S. aureus* NCTCΔ*atl* mutant and its parental strain NCTC8325-4 were obtained by harvesting overnight cultures at 10,000×*g* for 10 min at 4°C, and filtering the supernatants through a 0.22 μm filter (Merck Millipore). Sterility was confirmed by plating the super-natants on TSA plates at 30°C. To image the effect of the secreted Atl products on the PGRP-SA binding, exponentially growing NCTCΔ*atl* and NCTC8325-4 cells (500 μl at OD 0.5) were incubated with the previously isolated cell-free supernatants (500 μl) for 30 min at 30°C. Cells were then washed twice with cold PBS, incubated with mCherry_PGRP-SA, and imaged by fluorescence microscopy.

To quantify mCherry_PGRP-SA binding to the parental and the *atl* null mutant *S. aureus* cells pre-treated with secreted Atl products, the cultures were grown to mid-exponential phase (OD ~0.5), centrifuged, and resuspended in TSB to an OD of ~2.5. A 100 μl volume of each culture was placed into the wells of a 96-well plate and harvested at RT, 5 min at 3600×*g*. Cells were washed once with 100 μl of PBS and then resuspended in 100 μl of cell-free supernatants (prepared as described above). After 30 min of incubation at 30°C, the cells were washed twice with 100 μl of PBS. Quantification of mCherry_PGRP-SA binding was done as described above.

## Effect of the activities of AM amidase, GL glucosaminidase, LytA amidase, and mutanolysin muramidase on mCherry_PGRP-SA binding to the *S. aureus* bacterial surface

AM (15.6 and 390.1 nM), GL (18.7 and 467.4 nM), AM plus GL (15.6 and 18.7 nM, respectively), LytA (3.5 nM), and mutanolysin (108.7 nM) were added to 200 μl of PBS-washed NCTCΔ*atl* cells col-lected during exponential growth (500 μl at OD 0.5). After 30 min incubation at 30°C, cells were washed twice with cold PBS, incubated with mCherry_PGRP-SA as described before, and imaged by fluorescent microscopy.

## Dot-blot analysis of AM and GL bound to *S. aureus* cells

To determine the relative amount of AM and GL still associated with the bacterial cells following the treatment with these two enzymes, cells were treated with AM and GL as described above and washed three times with PBS (control pellets without the enzymes, without washing, or without cells were used). Samples were serially diluted 1:2 to concentrations ranging from 6 μg/ml (undiluted) to 0.37 μg/ml (lowest dilution), and 2 μl were spotted onto a polyvinylidene difluoride (PVDF) membrane (GE Healthcare Life Sciences) and immunostained with specific anti-His mouse antibodies. Washed, unwashed, and control bacterial cells were also labeled with mCherry_PGRP-SA as described above and observed by fluorescence microscopy.

## Fluorescence microscopy and image acquisition

Bacterial cells samples (2 μl) incubated with fluorescent tagged mCherry_PGRP-SA were placed on a thin layer of 1% agarose (Bio-Rad) in PBS. For time-lapse microscopy experiments, 2 μl of NCTCΔ*atl* culture, pre-labeled with mCherry-PGRP-SA, were placed on a thin layer of 0.6% agarose contain-ing filtered supernatants of either NCTC8325-4 or NCTCΔ*atl* cells. After 2.5 min, phase contrast and fluorescence images were taken every 30 s for 20.5 min and a final image was acquired after 50 min of incubation. Imaging was performed using a Leica DM 6000B microscope equipped with an iXon$^{EM}$ + EMCCD camera (Andor Technologies), using Metamorph software (Meta Imaging series 7.5). Images were analyzed using ImageJ software (*Abràmoff et al., 2004*).

## Fly survival

Overnight cultures of bacteria (10 ml) were washed and resuspended in PBS so that their final $OD_{600}$ was 0.350. Healthy looking adult female flies (n = 25), 2–4 days old, were injected in the thorax with 32 nl of a bacterial cell suspension (approximately 100 CFUs) or with PBS, using a nanoinjector (Nanoject II; Drummond Scientific, Broomall, PA), and fly survival was determined as previously described (*Atilano et al., 2011*). In the fly survival assay for the pathogenicity rescue of *atl* null mutants, 100 μl of each

bacterial cell suspension were incubated with cell-free supernatants of wild type or Δ*atl* mutants (prepared as described above) for 30 min at 30°C. Treated bacterial cells were washed twice, resuspended in an equal volume of PBS and further diluted so that approximately 100 CFUs would be injected per fly. Each experiment was performed in triplicate, on different days. Following injection, flies were kept at 30°C and survival assessed every 12 h over a period of 3 days. Since the trends in survival were the same for each sample of a triplicate (i.e., survival curves were positioned similarly, relative to one another), the three independent biological repeats were combined (n = 75) and estimates of survival curves were plotted.

Fly survival was also tested using PGRP-SA pre-coated *S. aureus* cells. For that purpose, overnight NCTCΔ*atl* bacterial cells were washed and resuspended in PBS (200 µl; $OD_{600}$ 0.350) and pre-coated with mCherry_PGRP-SA at 300 µg/ml for 5 min at 4°C. Cells were washed once and resuspended in an equal volume of ice-cold PBS. PBS suspensions with non-coated *atl* cells or containing only mCherry_PGRP-SA (300 µg/ml) were used as experiment controls. Controls and samples were further diluted in PBS (1:250) and used to infect PGRP-SA deficient flies as described above.

The bacterial load per insect during infection was assessed as previously described (*Atilano et al., 2011*). Six female flies were homogenized in 1000 µl TSB with a micropestle device (VWR, Radnor, PA) until pieces of tissue were no longer visible. Homogenate samples (50 µl) underwent a 10-fold dilution series from 1× to 1/100,000× before being spread onto TSB agar plates and onto mannitol salt agar. The number of CFUs was scored after 24 h at 30°C. Triplicate assays were performed.

## Measurements of *Drs* expression levels

Total RNA was extracted from homogenized female flies (n = 6) using the Total RNA Purification Plus Kit (Norgen, Canada) according to the manufacturers' instructions. A Maxima First Strand cDNA Synthesis Kit (Thermo Scientific) was used to produce cDNA. The total RNA (500 ng) was mixed with 5× reaction mix buffer (4 µl), Maxima enzyme mix (2 µl), and RNase-free $dH_2O$ to a final volume of 20 µl. cDNA was synthesized in a T100 Thermal Cycler (Bio-Rad) and stored at −20°C. Drosomycin (FlyBase annotation symbol: CG10810) and tbp (FlyBase annotation symbol: CG9874) expression levels were measured using primer pairs: Drs (+) and Drs (−); Tbp (+) and Tbp (−). The housekeeping gene *tbp* (*Matta et al., 2011*) was used as a control to normalize the expression of the *Drs* gene. qPCR reactions were performed using the SensiFast SYBR No-ROX Kit (Bioline, UK) as outlined in the manufacturer's instructions, in a Rotor-Gene Q real-time PCR cycler with a 72-well rotor (Qiagen). Negative controls had an equivalent amount of total RNA or no cDNA template. Three biological repeats were performed per time point. Gene expression was calculated on the basis of the comparative threshold cycle ($C_T$) value (*Schmittgen and Livak, 2008*). Levels of gene expression in all groups were shown as a ratio to the control group value.

## Statistical data analysis

Data for the mCherry_PGRP-SA binding to bacteria assays (n = 10–50) was non-normal but with equal variance, and therefore a non-parametric Kruskal–Wallis test followed by Dunn's multiple comparison was applied. Estimated survival curves were constructed from the raw data sets and the log-rank (Mantel–Cox) test used to determine statistical significance between the curves. For clarity, 95% confidence intervals have been omitted from the graphs.

When the bacterial CFU numbers in fly survival assays did not present a normal distribution (Lilliefors test, p<0.05), log10 transformation was used. Repeated measures two-way ANOVA was used to analyze significant differences over time and between bacterial strains. Bonferroni post-tests were used to locate the time points where mean values were statistically separable between the two bacterial strains.

Analysis for qPCR data showed that the data followed a normal distribution (Lilliefors test, p>0.1) and had equal variance (Levene's test, p>0.1). Thus, mean values were plotted with 95% confidence and repeated measures two-way ANOVA was used to look for significant differences between bacterial wild type strain and *atl* null mutant over time. Bonferroni post-tests were used to locate the time points where mean values were statistically separable between the two bacterial strains.

All data were plotted and analyzed using GraphPad Prism 6 (GraphPad Software).

## Acknowledgements

This study was funded by Fundação para a Ciência e Tecnologia (FCT), Lisbon, Portugal, through research grants PTDC/SAU-IMU/111806/2009 and PTDC/BIA-MIC/111817/2009 (SRF),

PTDC/BIA-BCM/099152/2008 (MGP), PTDC/BIA-MIC/101375/2008 (RGS) and PEst-OE/EQB/LA0004/2011. Wellcome Trust funded grant WT087680 (PL). ERC funded grant ERC-2012-StG-310987 (MGP). MLA, PMP, FV, MJC, PR, and IRG were supported by FCT fellowships SFRH/BD/28440/2006, SFRH/BD/41119/2007, SFRH/BD/78748/2011, SFRH/BPD/77758/2011, SFRH/BPD/23812/2005, and SFRH/BD/70162/2010, respectively.

## Additional information

### Funding

| Funder | Grant reference number | Author |
|---|---|---|
| Fundação para a Ciência e Tecnologia | PTDC/SAU-IMU/111806/2009 | Sérgio Raposo Filipe |
| Wellcome Trust | WT087680 | Petros Ligoxygakis |
| European Research Council | ERC-2012-StG-310987 | Mariana Gomes Pinho |
| Fundação para a Ciência e Tecnologia | PTDC/BIA-MIC/111817/2009 | Sérgio Raposo Filipe |
| Fundação para a Ciência e Tecnologia | PTDC/BIA-BCM/099152/2008 | Mariana Gomes Pinho |
| Fundação para a Ciência e Tecnologia | PTDC/BIA-MIC/101375/2008 | Rita Gonçalves Sobral |
| Fundação para a Ciência e Tecnologia | PEst-OE/EQB/LA0004/2011 | Mariana Gomes Pinho, Sérgio Raposo Filipe |
| Fundação para a Ciência e Tecnologia | Fellowship SFRH/BD/28440/2006 | Magda Luciana Atilano |
| Fundação para a Ciência e Tecnologia | Fellowship SFRH/BD/41119/2007 | Pedro Matos Pereira |
| Fundação para a Ciência e Tecnologia | Fellowship SFRH/BD/78748/2011 | Filipa Vaz |
| Fundação para a Ciência e Tecnologia | Fellowship SFRH/BD/77758/2011 | Maria João Catalão |
| Fundação para a Ciência e Tecnologia | Fellowship SFRH/BD/23812/2005 | Patricia Reed |
| Fundação para a Ciência e Tecnologia | Fellowship SFRH/BD/70162/2010 | Inês Ramos Grilo |

The funders had no role in study design, data collection and interpretation, or the decision to submit the work for publication.

### Author contributions

MLA, PMP, Conception and design, Acquisition of data, Analysis and interpretation of data, Drafting or revising the article, Contributed unpublished essential data or reagents; FV, MJC, Acquisition of data, Analysis and interpretation of data, Contributed unpublished essential data or reagents; PR, Drafting or revising the article, Contributed unpublished essential data or reagents; IRG, RGS, Acquisition of data, Contributed unpublished essential data or reagents; PL, Conception and design, Analysis and interpretation of data, Drafting or revising the article; MGP, SRF, Conception and design, Analysis and interpretation of data, Drafting or revising the article, Contributed unpublished essential data or reagents

## Additional files

### Supplementary files

• Supplementary file 1. Strains and plasmids used in this study.

• Supplementary file 2. Primers used in this study.

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
