## [Decision Letter]

[Editors’ note: a previous version of this study was rejected after peer review, but the authors submitted for reconsideration. The two decision letters after peer review are shown below.]

Thank you for choosing to send your work entitled “Bacterial autolysins trim cell surface peptidoglycan to prevent detection by the Drosophila innate immune system” for consideration at *eLife*. Your full submission has been evaluated by a Senior editor and 3 peer reviewers. One of them was Roberto Kolter, who is a member of our Board of Reviewing Editors. One of two remaining reviewers was Bruno Lamaitre. The third wished to remain anonymous. They reached their consensus decision after discussions amongst them.

In short, they have decided to reject the submission in its present form. They felt that there was still significant amount of experimental work to be done to fully support your interpretation that the autolysin Atl is serving to mask *S. aureus* and *S. pneumonia* from Drosophila's innate immune system.

However, they felt that if the additional work would indeed support this main tenet of your conclusions it would be of great interest. Thus, they would be willing to consider a new submission. Specifically, these are the key experiments that they saw lacking in the current work:

1) Complement PGRP-SA null flies with an *atl* mutant strain pre-coated with PGRP-SA.

2) Use sub-inhibitory concentrations of mutanolysin to trim the outer layer of peptidoglycan and then determine if that abolishes PGRP-SA binding to the *atl* mutant

3) Infect with a heat-killed *atl* mutant to determine if that protects the flies from infection with wild-type.

4) Check that the AM or GL mutated proteins are indeed deficient for the enzymatic activity.

5) The levels of Toll pathway activity should be monitored upon injection of wt and *atl* mutant

6) The persistence of the wt and *atl S. aureus* should be monitored over time

7) Performing survival experiments using the yw DD as a wild-type fly or use appropriate control

*Reviewer 1*
*(Roberto Kolter):*

In general I liked the “bottom line” results of this paper: that an autolysin has the unexpected function of “trimming” peptidoglycan part that would otherwise be exposed on the cell surface. In this way, the cell can better evade the fly's innate immunity. The authors not only show the effect of autolysin mutants on binding by purified fly protein, they also work with purified autolysin and fragments. Importantly, they show effect on adult flies and they extend the findings to Streptococcus. From the perspective of an outsider (I am no expert in cell wall nor innate immunity) these results do appear to have wide implications. However, I did find that the paper could use some polishing in the written style.

*Reviewer*
*2:*

Atilano et al present an elegant study on the role of autolysins in escaping innate immune response. The authors extend their observations from their model organism *Staphylococcus aureus* to *Streptococcus pneumonia* and present a model for their results. The model proposes that autolysins trim the exposed peptidoglycan to remove any potential binding sites for innate immune receptors of the PGRP family, in this particular case, PGRP-SA. The authors conclude that it is rather binding to the exposed peptidoglycan than detection of release peptidoglycan that is important for host response. I do not agree with their conclusion and model. The authors can only conclude that autolytic activities such as amidases are important for innate immune escape (not novel) but not on the structural nature of the peptidoglycan relevant for detection in vivo (soluble versus surface attached). PGRPs such as PGRP-SA require the stem peptide for detection. Hence, amidase activities are bound to reduce the biological activity of the peptidoglycan, both of the attached as the release one. Amidase treated peptidoglycan or peptidoglycan fragments are no longer ligands for PGRP-SA due to the absolute requirement of both the stem peptide and the sugar backbone for binding. In fact, the Drosophila produces also amidases to dampen the biological activity of circulating peptidoglycan such as PGRP-LB. To conclude that it is PGRP-SA binding to the outer layers of the peptidoglycan rather soluble circulating peptidoglycan fragments that are important for innate immune detection by PGRP-SA, the authors should perform a series of experiments:

First, the authors should functionally complement PGRP-SA null flies with *atl* mutant strain pre-coated with PGRP-SA. The authors would have to demonstrate that the same amount of circulating recombinant PGRP-SA is unable to initiate alone a host response and that none of the pre-coated PGRP-SA is released into circulation from the bacterial surface during the period of the experiment. Otherwise, the authors should moderate their conclusions on the nature of the peptidoglycan that is important for PGRP-SA mediate response.

Second, if their model is correct, the trimming of the outer layers of the peptidoglycan by sub-inhibitory concentrations of mutanolysin should abolish PGRP-SA binding to the *atl* mutant without affecting the biological activity of the released peptidoglycan. The supernatants should induce a PGRP-SA-dependent host response, while the *atl* mutant would see its virulence restored in vivo because it would not be able to bind circulating PGRP-SA on its surface.

Third, if surface binding alone is important, then co-infection of lethal doses of the wild type *S. aureus* strain should be countered by the host response triggered by co-administration of the heat-killed *atl* mutant. The heat-killed *alt* mutant should recruit to the surface PGRP-SA triggering a host response able to coop with the wild type strain. By definition, heat-killed *atl* mutant should not release circulating peptidoglycan at all.

Fourth, the authors should confirm that the mutations introduced in Atl to inactivate the amidase and the glucosaminidase activity do indeed abolish their activity (zymogram assay for example). More importantly, the authors must provide evidence that these mutations do not create overall protein misfolding in which case the authors can not conclude that it is the two activities that are required, only the entire protein. Atl has the R1, R2 and R3 domains that besides mediating Atl binding to the bacteria cell surface, they mediate also interactions with matrix proteins or biofilm formation, among other properties. If the protein fold is maintained, these properties that are independent of the enzymatic activity should not change.

*Reviewer*
*3:*

*Staphylococcus aureus* is a highly pathogenic bacterium to many hosts including Drosophila and mammals. The Filipe group analyzes the mechanism allowing this bacterium to be hidden from pattern-recognition detection. They have previously shown that teichoic acid protects *S. aureus* from being detected by PGRP-SA, a secreted pattern-recognition receptor of Drosophila that interacts with peptidoglycan. In this study Atilano et al. report the identification of one gene, *atl*, encoding an autolysin that reduces the binding of PGRP-SA to *S. aureus*. They also show that the *S. aureus atl* mutant is less virulent towards Drosophila in vivo (as previously reported by another paper, Tabuchi et al.). They also show that both teichoic acid and Atl synergistically reduce the immunogenicity of *S. aureus*. Based on their data, they propose a model in which the autolysin Atl trims peptidoglycans, reducing its exposure to secreted pattern-recognition receptors. To my knowledge, this work uncovers a new mechanism for autolysin in preventing the detection of Gram-positive bacteria by host peptidoglycan sensors. The paper is a very interesting and contains many striking results. However, it still lacks critical experiments to clearly establish that Atl affects Toll activation by *S. aureus* in vivo.

1) The authors claim that Atl trims peptidoglycan (PGN) from bacterial surfaces but do not provide any direct evidence for that. They even say that PGN from the *atl* mutant does not differ from wild-type PGN. Is there a way of monitoring PGN exposed at the bacterial surface? If not, I might be a bit more cautious in the wording saying that their data are compatible with a model in which Atl trims PGN, thereby reducing PGRP recognition.

2) In vivo analysis

This is a key aspect of the paper. All the previous experiments nicely show that Atl contributes to conceal PGN from PGRP-SA binding. This is however not fully unexpected because Atl has PGN hydrolysis activity. The important point is therefore to know if Atl does indeed contribute to conceal *S. aureus* from Toll activation in vivo. More work should be performed to substantiate this point by adding the following experiments:

- Experiments in Figure 8 show that Atl contributes to *S. aureus* virulence but they do not show if this is linked to a higher activation of the Toll pathway. The levels of Toll pathway activity should be monitored upon injection of wt and *atl S. aureus*. (It is possible that the effect on PGRP-SA is masked by the activation of the psh pathway by proteases released from *S. aureus*. In this case, Toll pathway activity should be monitored in *psh* mutants upon injection wt and *atl S. aureus*.) Those experiences are easy to perform.

- The persistence of the wt and *atl S. aureus* (at 2-3 times points) should be monitored. This will tell us whether the lower pathogenicity correlates with a lower proliferation rate of the bacterium. This experiment is also essential to understand how Atl influences Toll pathway activity (point 2b)

3) Results concerning the synergistic action of TagO and Atl should be moved to the Results section rather than briefly mentioned in the Discussion. Survival analysis of flies infected with simple and double mutants could be presented as well as persistence and Toll pathway activation.

4) To my knowledge, PGRP-SA (*seml*) is a mutation that was generated in the yw, DD background. Therefore, I question the notion that the wild-type strain used in the study (Isogenic *Drosophila* Bloomington#25174) is isogenic to the PGRP-SA mutant stock. If I am right, I would recommend performing survival experiments using the yw DD as a wild-type. Note that the use of the yw background is far from being ideal (the y mutation tends to make flies more susceptible to infection). Ideally, survival should be performed in y+,w+ background (easy to do by comparing wild-type with ywDD,PGRP-SA/Def(PGRP-SA).

[Editors’ note: what now follows is the decision letter after the authors submitted for further consideration.]

Thank you for resubmitting your work entitled “Bacterial autolysins trim cell surface peptidoglycan to prevent detection by the Drosophila innate immune system” for further consideration at *eLife*. Your revised article has been favorably evaluated by a Senior editor, a member of the Board of Reviewing Editors (Roberto Kolter) and 2 Reviewers. The manuscript is much improved but there are some remaining issues that need to be addressed before acceptance.

1) Authors were asked to perform the trimming assay with mutanolysin. The authors indicate this was done and worked just like Atl. This is an important result to support their model but they decided to leave it out of the manuscript. The reviewers feel the result should be included in the manuscript.

2) There might be many reasons for the heat-killed experiment not working and that Toll activation alone might not be sufficient. Indeed, phagocytosis might be more important than Toll activation and antimicrobial peptide production. Thus, binding of PGRP-SA would be essential and the wild-type *S. aureus* would still escape. However, the Toll activation should be restored. Is that the case?

3) The authors have indicated that they have performed statistical analysis of their experiments and added it to the figure legends. In some figures, indeed, the statistics is immediately apparent with the presence of asterisks. However, in all the fly experiments, this information is difficult to find. The authors should add the same information in the figures for the fly experiments. In particular, Figure 7 is of importance because if the model is correct the addition of PGRP-SA to *alt* mutants should restore survival. There is an increased survival in time but not in absolute numbers. I could not find in the text if this difference is statistically significant or not. Importantly, the addition of PGRP-SA does not restore fly survival to the levels of the parental background (Figure 6). First, the authors should indicate whether it is statistically different or not and comment on the eventual (or not if not significantly different) partial complementation.

4) It would have been better to extend the persistence analysis in Figure 5 (as in Figure 5). The effect of the *atl* deficiency on Drosomycin activation is modest and observed only at the 12 h time point. Why? A possible explanation is that *S. aureus* can also activate the Toll pathway by the protease route and that PGRP-SA is not be only important for Toll pathway activation but also for inducing the melanization cascade (see first figure of the Bishoff PGRP-SD paper). Would it be possible to analyze the effect of Atl in *S. aureus* that do not divide? It is true that these experiments are not easy to perform. If this is the case, that fact should at least be discussed.

---

## [Author Response]

[Editors’ note: the author responses to the first round of peer review follow.]

*1) Complement PGRP-SA null flies with an* atl *mutant strain pre-coated with PGRP-SA.*

Successfully done: see Answer 3.

*2) Use sub-inhibitory concentrations of mutanolysin to trim the outer layer of peptidoglycan and then determine if that abolishes PGRP-SA binding to the* atl *mutant*

Successfully done: see Answer 4.

*3) Infect with a heat-killed* atl *mutant to determine if that protects the flies from infection with wild-type*.

Done, but the presence of heat killed did not protect flies from infection with wild type.

See explanation in Answer 5 and alternative experiment described in Answer 2.

*4) Check that the AM or GL mutated proteins are indeed deficient for the enzymatic activity*.

Successfully done: see Answer 6.

*5) The levels of Toll pathway activity should be monitored upon injection of wt and* atl *mutant.*

Successfully done: see Answer 8.

*6) The persistence of the wt and* atl S. aureus *should be monitored over time.*

Successfully done: see Answer 9.

*7) Performing survival experiments using the yw DD as a wild-type fly or use appropriate control*.

Successfully done: see Answer 11.

Reviewer 1 (Roberto Kolter):

*In general I liked the “bottom line” results of this paper: that an autolysin has the unexpected function of “trimming” peptidoglycan part that would otherwise be exposed on the cell surface. In this way, the cell can better evade the fly's innate immunity. The authors not only show the effect of autolysin mutants on binding by purified fly protein, they also work with purified autolysin and fragments. Importantly, they show effect on adult flies and they extend the findings to Streptococcus. From the perspective of an outsider (I am no expert in cell wall nor innate immunity) these results do appear to have wide implications. However, I did find that the paper could use some polishing in the written style*.

**Answer 1**. All suggested changes were made in the manuscript.

Reviewer 2:

*Atilano et al present an elegant study on the role of autolysins in escaping innate immune response. The authors extend their observations from their model organism* Staphylococcus aureus *to* Streptococcus pneumonia *and present a model for their results. The model proposes that autolysins trim the exposed peptidoglycan to remove any potential binding sites for innate immune receptors of the PGRP family, in this particular case, PGRP-SA. The authors conclude that it is rather binding to the exposed peptidoglycan than detection of release peptidoglycan that is important for host response. I do not agree with their conclusion and model. The authors can only conclude that autolytic activities such as amidases are important for innate immune escape (not novel) but not on the structural nature of the peptidoglycan relevant for detection* in vivo *(soluble versus surface attached). PGRPs such as PGRP-SA require the stem peptide for detection. Hence, amidase activities are bound to reduce the biological activity of the peptidoglycan, both of the attached as the release one. Amidase treated peptidoglycan or peptidoglycan fragments are no longer ligands for PGRP-SA due to the absolute requirement of both the stem peptide and the sugar backbone for binding. In fact, the Drosophila produces also amidases to dampen the biological activity of circulating peptidoglycan such as PGRP-LB*.

**Answer 2.** We thank the reviewer for making an important point regarding two possible mechanisms for the role of Atl, which indeed were valid and we had not addressed in the original manuscript.

The second paragraph in the Results section entitled “Trimming of the bacterial cell surface by Atl PGN hydrolytic activities recovers the ability of *S. aureus atl* mutant bacteria to kill infected Drosophila” now clearly state two non-mutually exclusive hypotheses for the decreased virulence of the *atl* mutant:

Hypothesis 1: *S. aureus* wild type bacteria produce amidase that cleaves the stem peptide moiety of soluble PGN fragments, which is required for binding of PGRPs to PGN, therefore reducing the inflammatory activity of the soluble PGN. However, the *atl* mutant would release intact muropeptides, easily detected by PGRP-SA, and therefore would be unable to evade detection by the innate immune system;

Hypothesis 2: *S. aureus* wild type bacteria shave and remove accessible PGN fragments at the surface of bacteria, eliminating the binding sites for PGRPs. However, the *atl* mutant would have extending PGN fragments at its surface, which would be easily detected by PGRP-SA, inducing an immune response.

We performed the experiments suggested by the reviewer (see below) to address this question, but we also tried to come up with a more direct way of distinguishing the two hypotheses. We reasoned that if the first hypothesis was predominantly correct, inactivation of the amidase activity, but not of the glucosaminidase (which cleaves the glycans and therefore releases intact, inflammatory PGN fragments), should result in decreased virulence. Alternatively, if the second hypothesis was predominantly correct, as favored in our manuscript, inactivation of either the amidase or the glucosaminidase activity should result in decreased virulence, as both activities shave detectable PGN (containing peptides) from the bacteria surface.

We therefore infected flies with bacteria with impaired amidase activity (AM^H265A^) or impaired glucosaminidase activity (GLE^1128A^) and observed that both were impaired in their ability to kill flies, similarly to the *atl* null mutant, in accordance with the second hypothesis (Figure 7). Although the effect of Atl on soluble PGN fragments may have a role, we think this experiment clearly supports our original model.

*To conclude that it is PGRP-SA binding to the outer layers of the peptidoglycan rather soluble circulating peptidoglycan fragments that are important for innate immune detection by PGRP-SA, the authors should perform a series of*
*experiments*:

*First, the authors should functionally complement PGRP-SA null flies with* atl *mutant strain pre-coated with PGRP-SA. The authors would have to demonstrate that the same amount of circulating recombinant PGRP-SA is unable to initiate alone a host response and that none of the pre-coated PGRP-SA is released into circulation from the bacterial surface during the period of the experiment. Otherwise, the authors should moderate their conclusions on the nature of the peptidoglycan that is important for PGRP-SA mediate response*.

**Answer 3.** As suggested, we have complemented PGRP-SA null flies with the *atl* mutant strain pre-coated with PGRP-SA. In accordance to the reviewer prediction, *seml* flies infected with these pre-coated bacteria were capable of resisting better the bacterial infection, when compared with flies infected with PGRP-SA uncoated bacteria, as shown in Figure 7. Although we know that PGRP-SA stays bound to the surface (see timelapse in Figure 9), we cannot determine the amount of the pre-coated PGRP-SA released into the haemolymph during the time of the experiment.

*Second, if their model is correct, the trimming of the outer layers of the peptidoglycan by sub-inhibitory concentrations of mutanolysin should abolish PGRP-SA binding to the* atl *mutant without affecting the biological activity of the released peptidoglycan. The supernatants should induce a PGRP-SA-dependent host response, while the* atl *mutant would see its virulence restored* in vivo *because it would not be able to bind circulating PGRP-SA on its surface*.

**Answer 4.** As suggested we have used mutanolysin to shave surface peptidoglycan. As expected PGRP-SA was unable to bind bacterial *atl* mutant cells treated with mutanolysin. Given the large number of figures already included in the manuscript, we did not include these results, as mutanolysin has the same activity as glucosaminidase, whose effect is shown Figure 4.

*Third, if surface binding alone is important, then co-infection of lethal doses of the wild type* S. aureus *strain should be countered by the host response triggered by co-administration of the heat-killed* atl *mutant. The heat-killed* alt *mutant should recruit to the surface PGRP-SA triggering a host response able to coop with the wild type strain. By definition, heat-killed* atl *mutant should not release circulating peptidoglycan at all*.

**Answer 5.** We have done the suggested experiment but co-administration of wild type bacteria and heat-killed *atl* mutant cells, did not have an observed effect. A possibility is that PGRP-SA binding to a specific bacterium, not just the activation of the Toll pathway, is required for its elimination by the innate immune response.

*Fourth, the authors should confirm that the mutations introduced in Atl to inactivate the amidase and the glucosaminidase activity do indeed abolish their activity (zymogram assay for example). More importantly, the authors must provide evidence that these mutations do not create overall protein misfolding in which case the authors can not conclude that it is the two activities that are required, only the entire protein. Atl has the R1, R2 and R3 domains that besides mediating Atl binding to the bacteria cell surface, they mediate also interactions with matrix proteins or biofilm formation, among other properties. If the protein fold is maintained, these properties that are independent of the enzymatic activity should not change*.

**Answer 6.** We have performed the suggested zymogram and shown that mutations introduced in Atl to inactivate the amidase and the glucosaminidase activity do indeed abolish only the targeted activity, as seen in Figure 3.

Reviewer 3:

Staphylococcus aureus *is a highly pathogenic bacterium to many hosts including Drosophila and mammals. The Filipe group analyzes the mechanism allowing this bacterium to be hidden from pattern-recognition detection. They have previously shown that teichoic acid protects* S. aureus *from being detected by PGRP-SA, a secreted pattern-recognition receptor of Drosophila that interacts with peptidoglycan. In this study Atilano et al. report the identification of one gene,* atl, *encoding an autolysin that reduces the binding of PGRP-SA to* S. aureus. *They also show that the* S. aureus atl *mutant is less virulent towards Drosophila* in vivo *(as previously reported by another paper, Tabuchi et al.). They also show that both teichoic acid and Atl synergistically reduce the immunogenicity of* S. aureus*. Based on their data, they propose a model in which the autolysin Atl trims peptidoglycans, reducing its exposure to secreted pattern-recognition receptors. To my knowledge, this work uncovers a new mechanism for autolysin in preventing the detection of Gram-positive bacteria by host peptidoglycan sensors. The paper is a very interesting and contains many striking results. However, it still lacks critical experiments to clearly establish that Atl affects Toll activation by* S. aureus in vivo.

*1) The authors claim that Atl trims peptidoglycan (PGN) from bacterial surfaces but do not provide any direct evidence for that. They even say that PGN from the* atl *mutant does not differ from wild-type PGN. Is there a way of monitoring PGN exposed at the bacterial surface? If not, I might be a bit more cautious in the wording saying that their data are compatible with a model in which Atl trims PGN, thereby reducing PGRP recognition*.

**Answer 7.** We cannot directly monitor PGN exposed at the bacterial surface. As suggested by the reviewer, we have rephrase the text that stated that PGN from *atl* mutants does not differ from wild-type PGN in order to highlight that they had the same muropeptide PGN composition (second paragraph of the Results section) and that Atl presumably release PGN fragments (final paragraph of the section entitled “Trimming of the bacterial cell surface by Atl PGN hydrolytic activities recovers the ability of *S. aureus atl* mutant bacteria to kill infected Drosophila”.

*2) In vivo*
*analysis*

*This is a key aspect of the paper. All the previous experiments nicely show that Atl contributes to conceal PGN from PGRP-SA binding. This is however not fully unexpected because Atl has PGN hydrolysis activity. The important point is therefore to know if Atl does indeed contribute to conceal* S. aureus *from Toll activation* in vivo*. More work should be performed to substantiate this point by adding the following experiments:*

*- Experiments in*
Figure 8
*show that Atl contributes to* S. aureus *virulence but they do not show if this is linked to a higher activation of the Toll pathway. The levels of Toll pathway activity should be monitored upon injection of wt and* atl S. aureus*. (It is possible that the effect on PGRP-SA is masked by the activation of the psh pathway by proteases released from* S. aureus*. In this case, Toll pathway activity should be monitored in* psh *mutants upon injection wt and* atl S. aureus*.) Those experiences are easy to perform*.

**Answer 8.** We have monitored the levels of the Toll pathway using drosomycin expression as a reporter. When the level of drosomycin expression was normalized for the number of bacteria present in infected flies at each particular time point, the activation of the drosomycin expression was higher for the *atl* mutant than for the wild type *S. aureus* strain at 12 h post-infection (Figure 6).

*- The persistence of the wt and* atl S. aureus *(at 2-3 times points) should be monitored. This will tell us whether the lower pathogenicity correlates with a lower proliferation rate of the bacterium. This experiment is also essential to understand how Atl influences Toll pathway activity (point 2b)*.

**Answer 9.** We have monitored the persistence of wt and *atl S. aureus* strains in the *seml* flies and found identical number of the two bacterial strains (Figure 6), showing that the *atl* mutant is able to grow well in Drosophila flies. Even when injected into wild type flies, the *atl S. aureus* cells are able to grow as well as the wild type during the initial period of infection (6 h), before the activation of drosomycin is observed and *atl* mutant bacteria start being eliminated by the immune system (Figure 6)

*3) Results concerning the synergistic action of TagO and Atl should be moved to the Results section rather than briefly mentioned in the Discussion. Survival analysis of flies infected with simple and double mutants could be presented as well as persistence and Toll pathway activation*.

**Answer 10**. As requested, we have moved the results of the synergistic action of TagO and Atl to the results (new panel Figure 2). These experiments aimed only at showing that the role of Atl in PGRP-SA binding was different from that of TagO. We did not aim to characterize an Atl TagO double mutant in this manuscript and for that reason we did not include new data on the double mutant.

*4) To my knowledge, PGRP-SA* (seml) *is a mutation that was generated in the yw, DD background. Therefore, I question the notion that the wild-type strain used in the study (Isogenic* Drosophila *Bloomington#25174) is isogenic to the PGRP-SA mutant stock. If I am right, I would recommend performing survival experiments using the yw DD as a wild-type. Note that the use of the yw background is far from being ideal (the y mutation tends to make flies more susceptible to infection). Ideally, survival should be performed in y+,w+ background (easy to do by comparing wild-type*
*with ywDD,PGRP-SA/Def(PGRP-SA)*

**Answer 11**. As requested, we have performed the survival experiments using yw (Bloomington#6599) background. The results were similar to the ones previously obtained with the Bloomington#25174 and are shown in Figure 6.

[Editors’ note: the author responses to the re-review follow.]

*1) Authors were asked to perform the trimming assay with mutanolysin. The authors indicate this was done and worked just like Atl. This is an important result to support their model but they decided to leave it out of the manuscript. The reviewers feel the result should be included in the manuscript*.

These results are now shown in Figure 11 and these experiments are referred in the manuscript (at the end of the Results section).

*2) There might be many reasons for the heat-killed experiment not working and that Toll activation alone might not be sufficient. Indeed, phagocytosis might be more important than Toll activation and antimicrobial peptide production. Thus, binding of PGRP-SA would be essential and the wild-type* S. aureus *would still escape. However, the Toll activation should be restored. Is that the*
*case?*

The reviewer had initially asked us to test the following whether: “co-infection of lethal doses of the wild type *S. aureus* strain should be countered by the host response triggered by coadministration of the heat-killed *atl* mutant”.

Therefore the experiment done was co-infecting wt flies with wt *S. aureus* mixed with heat killed *atl* mutants cells. Given that the flies were wt, the Toll pathway was intact from the beginning. Given that wt *S. aureus* cells were injected, activation of the Toll pathways was expected and observed as shown in Figure 6.

Nevertheless, we agree with the reviewer that there might indeed be multiple reasons why the heat-killed experiment did not work. As mentioned by the reviewer one other such factor is phagocytosis. However, there is no link between PGRP-SA and phagocytosis as shown by Dominique Ferrandon and colleagues (Nehme et al PLoS One 6, e14743, 2011). We also don´t think opsonisation is happening here. In our PLoS Pathogens paper ([4], PLoS Pathog 7, e1002421) we described how during host defence against another *S. aureus* mutant (TagO) that binds more PGRP-SA, bacterial killing by the host is dependent on PGRP-SA but not on Toll/NF-kB. This shows that as long as there is good access to peptidoglycan for PGRP-SA to bind around the cell surface of the bacterium, bacteria can be eliminated quickly and so Toll signalling is dispensable. Evidence from other insects (especially the very detailed work on the beetle *Tenebrio molitor* by the Lee lab over a number of years – for a very nice review of their work see Park et al Adv Exp Med Biol 708, p163, 2010) point towards the assembly of a “killing” complex that has to do with melanisation, an insect-specific response resembling complement activation. Ligoxygakis and Atilano are actively pursuing this line of research but we believe that it is beyond the scope of the present study.

*3) The authors have indicated that they have performed statistical analysis of their experiments and added it to the figure legends. In some figures, indeed, the statistics is immediately apparent with the presence of asterisks. However, in all the fly experiments, this information is difficult to find. The authors should add the same information in the figures for the fly experiments. In particular,*
Figure 7
*is of importance because if the model is correct the addition of PGRP-SA to* alt *mutants should restore survival. There is an increased survival in time but not in absolute numbers. I could not find in the text if this difference is statistically significant or not. Importantly, the addition of PGRP-SA does not restore fly survival to the levels of the parental background (*Figure 6*). First, the authors should indicate whether it is statistically different or not and comment on the eventual (or not if not significantly different) partial complementation*.

The requested information was introduced in Figures 6, 7 and 8 and 10, which now show asterisks to indicate curves whose difference to wt is statistically significant. This information is also detailed in the corresponding legends

A comment on the partial complementation was introduced in the last paragraph of the Results section entitled “Trimming of the bacterial cell surface by Atl PGN hydrolytic activities recovers the ability of *S. aureus atl* mutant bacteria to kill infected Drosophila”.

*4) It would have been better to extend the persistence analysis in*
Figure 5
*(as in*
Figure 5*). The effect of the* atl *deficiency on Drosomycin activation is modest and observed only at the 12 h time point. Why? A possible explanation is that* S. aureus *can also activate the Toll pathway by the protease route and that PGRP-SA is not be only important for Toll pathway activation but also for inducing the melanization cascade (see first figure of the Bishoff PGRP-SD paper). Would it be possible to analyze the effect of Atl in* S. aureus *that do not divide? It is true that these experiments are not easy to perform. If this is the case, that fact should at least be discussed*.

The persistent analysis in Figure 5 was not extended because the flies died. As mentioned above we also believe that melanisation plays a role but that is beyond the scope of this paper as are the experiments proposed in relation to *S. aureus* non-dividing mutants. However, we have discussed the point on melanisation extensively in [4], PLoS Pathog 7, e1002421. Wild type *S. aureus* may activate Toll via the protease route. However, host killing of the *atl* (this work) and TagO ([4], PLoS Pathog 7, e1002421) mutants is PGRP-SA dependent but Toll-independent. The point we want to make is not about host killing of wild type *S. aureus*. The point we want to explore by generating these bacterial mutants is how host defense changes when the bacterial surface is changing. This reveals not only the requirement for concealing pathogen-associated molecules from the host but also how the host might compensate when bacteria try to evade recognition.